# A dynamical computational model of theta generation in hippocampal circuits to study theta-gamma oscillations during neurostimulation

**Nikolaos Vardalakis[1,2], Amélie Aussel[1,2,3], Nicolas P Rougier[1,2,3], Fabien B Wagner[1]***

[1]University of Bordeaux, CNRS, IMN, Bordeaux, France; [2]University of Bordeaux, INRIA, IMN, Bordeaux, France; [3]University of Bordeaux, CNRS, Bordeaux INP, Talence, France

**\*For correspondence:**
fabien.wagner@u-bordeaux.fr

**Competing interest:** The authors declare that no competing interests exist.

**Abstract** Neurostimulation of the hippocampal formation has shown promising results for modulating memory but the underlying mechanisms remain unclear. In particular, the effects on hippocampal theta-nested gamma oscillations and theta phase reset, which are both crucial for memory processes, are unknown. Moreover, these effects cannot be investigated using current computational models, which consider theta oscillations with a fixed amplitude and phase velocity. Here, we developed a novel computational model that includes the medial septum, represented as a set of abstract Kuramoto oscillators producing a dynamical theta rhythm with phase reset, and the hippocampal formation, composed of biophysically realistic neurons and able to generate theta-nested gamma oscillations under theta drive. We showed that, for theta inputs just below the threshold to induce self-sustained theta-nested gamma oscillations, a single stimulation pulse could switch the network behavior from non-oscillatory to a state producing sustained oscillations. Next, we demonstrated that, for a weaker theta input, pulse train stimulation at the theta frequency could transiently restore seemingly physiological oscillations. Importantly, the presence of phase reset influenced whether these two effects depended on the phase at which stimulation onset was delivered, which has practical implications for designing neurostimulation protocols that are triggered by the phase of ongoing theta oscillations. This novel model opens new avenues for studying the effects of neurostimulation on the hippocampal formation. Furthermore, our hybrid approach that combines different levels of abstraction could be extended in future work to other neural circuits that produce dynamical brain rhythms.

## eLife assessment

This study presents a computational model to explore how neurostimulation could impact hippocampal theta oscillations. The computational model combines a detailed physiologically realistic hippocampus model and an abstract theta oscillator. The study could provide **valuable** predictions on pathological changes in this network. The modelling is based on **convincing** approaches that could be improved with experimental validation in future experiments.

## Introduction

Neurostimulation methods have emerged as promising therapeutic modalities to restore neurological functions in a broad range of motor and cognitive disorders (*Gupta et al., 2023*). In the context of learning and memory, deep brain stimulation (DBS) of the entorhinal area or hippocampus has

been shown to either enhance (*Suthana et al., 2012*; *Suthana and Fried, 2014*; *Titiz et al., 2017*; *Jun et al., 2019*) or disrupt (*Jacobs et al., 2016*; *Goyal et al., 2018*; *Lozano et al., 2016*) memory encoding. These conflicting results may originate from differences in experimental protocols and from a poor understanding of their biophysical underpinnings. Among such mechanisms, the involvement of hippocampal theta oscillations (4–12 Hz) and their interactions with higher-frequency gamma oscillations (30–120 Hz) in memory-related processes has been reported in multiple studies (*Lisman et al., 2005*; *de Almeida et al., 2007*; *Lega et al., 2012*; *Lin et al., 2017*; *Malkov et al., 2022*; *Abbaspoor et al., 2023*). Moreover, the modulation of gamma oscillations by the phase of theta oscillations in hippocampal circuits, a phenomenon termed theta-gamma phase-amplitude coupling (PAC), correlates with the efficacy of memory encoding and retrieval (*Jensen and Colgin, 2007*; *Tort et al., 2009*; *Canolty and Knight, 2010*; *Axmacher et al., 2010*; *Fell and Axmacher, 2011*; *Lisman and Jensen, 2013*; *Lega et al., 2016*). Experimental and computational work on the coupling between oscillatory rhythms has indicated that it originates from different neural architectures and correlates with a range of behavioral and cognitive functions, enabling the long-range synchronization of cortical areas and facilitating multi-item encoding in the context of memory (*Hyafil et al., 2015*).

Neurostimulation protocols that affect these rhythms, such as theta burst stimulation (*Titiz et al., 2017*), have also been shown to optimally induce long-term potentiation (LTP) (*Larson and Munkácsy, 2015*). Another potential mechanism underlying the effect of hippocampal neurostimulation might be the reset of the phase of theta oscillations in response to exogenous inputs, such as a novel sensory input or a pulse of electrical stimulation applied to the fornix or perforant path (*Buño et al., 1978*; *Williams and Givens, 2003*). Theta phase reset is known to facilitate LTP (*McCartney et al., 2004*) and naturally occurs during both encoding and retrieval of associative memories (*Kota et al., 2020*). In this context, the design of a computational model that replicates memory-related theta-gamma oscillations and theta phase reset is of uttermost importance to investigate the effects of electrical stimulation on the hippocampal formation and possibly optimize neurostimulation protocols for memory improvement.

Models of memory-related theta-gamma oscillations in the hippocampal formation have been developed across different resolution levels, ranging from abstract mean-field approaches (*Traub et al., 1997*; *Onslow et al., 2014*; *Segneri et al., 2020*) to biophysically realistic conductance-based models (*Lundqvist et al., 2006*; *Herman et al., 2013*; *Aussel et al., 2018*). Neural masses, which represent the mean activity of a neuronal population, can generate gamma oscillations through reciprocal interactions between an excitatory and inhibitory population (*Traub et al., 1997*; *Onslow et al., 2014*), or even using a self-projecting inhibitory population (*Segneri et al., 2020*). Under excitatory oscillatory input at theta frequencies, these models are capable of generating theta-nested gamma oscillations. Similarly, these oscillations can be observed in more complex models of the hippocampal formation composed of single-compartment excitatory and inhibitory neurons connected through conductance-based synapses following the Hodgkin-Huxley formalism (*Hodgkin and Huxley, 1952*), and driven by a fixed oscillatory theta input (*Aussel et al., 2018*). Theta-nested gamma oscillations also appear in multi-compartment models of prefrontal cortex activity during memory retrieval (*Lundqvist et al., 2006*; *Herman et al., 2013*). Finally, several models have investigated the functional link between the hippocampal theta rhythm and memory processes, showing that encoding and retrieval occur at different phases within each theta cycle through phasic changes in neuronal activity and LTP (e.g. *Hasselmo et al., 2002*; *Cutsuridis et al., 2010*).

In terms of neurostimulation, most computational work has focused on the mechanisms underlying DBS of the basal ganglia for motor disorders such as Parkinson's disease (*Rubin and Terman, 2004*; *Pirini et al., 2009*; *Mina et al., 2013*; *Ebert et al., 2014*), peripheral nerve stimulation (*Rattay et al., 2003*; *Kipping and Nogueira, 2022*), spinal cord stimulation (*Rattay et al., 2000*; *Capogrosso et al., 2013*), or has remained generic (*Basu et al., 2018*). However, models investigating neurostimulation of hippocampal circuits are scarce (*Hendrickson et al., 2016*; *Bingham et al., 2018*) and do not take into account the effects on theta-gamma oscillations. For example, a detailed multicompartment model of the rat dentate gyrus was able to replicate experimentally recorded local field potentials induced by different electrode placements and pulse amplitudes during stimulation of the perforant path (*Bingham et al., 2018*). To model the impact of neurostimulation on neuronal oscillations, a more abstract formalism based on Kuramoto phase oscillators (*Kuramoto, 1984*) has been introduced in the context of Parkinson's disease and essential tremor, which enabled the design of novel

neurostimulation paradigms that enhance or disrupt neuronal synchrony in basal ganglia circuits (*Tass, 2003*; *Ebert et al., 2014*; *Asllani et al., 2018*; *Weerasinghe et al., 2019*).

To our knowledge, there is currently no model of the hippocampal formation that is able to replicate both theta-gamma PAC and theta phase reset during neurostimulation. To investigate the effects of neurostimulation on hippocampal circuits while taking into account these two mechanisms, we modified an existing biophysical model of the hippocampal formation (*Aussel et al., 2018*). In this original model, however, the theta rhythm was considered as an oscillatory input of fixed amplitude and phase velocity, which is inconsistent with theta phase reset. To circumvent this limitation, we combined this biophysical model with abstract Kuramoto oscillators that acted as a dynamical source of theta rhythm, thereby modeling medial septum inputs to the hippocampal formation. This new hybrid dynamical model could generate both theta-nested gamma oscillations and theta phase reset, following a particular phase response curve (PRC) inspired by experimental literature (*Lengyel et al., 2005*; *Akam et al., 2012*; *Torben-Nielsen et al., 2010*).

We then leveraged this model to explore the effect of single-pulse and pulse-train stimulation on theta-gamma oscillations. In the absence of theta input from the medial septum, single-pulse stimulation produced a transient effect consisting of one or several bursts of activity, depending on stimulation amplitude. The presence of multiple bursts depended on the single-cell calcium dynamics and M-type potassium adaptation current. In the presence of weak theta input, designed to mimic a pathological state that impairs theta-gamma oscillations, single-pulse stimulation could produce long-lasting or even persistent activity by switching the network to a highly synchronized state characterized by theta-nested gamma oscillations. When phase reset was not included in the model, this effect was more pronounced when the stimulation pulse was delivered at the peak of the theta rhythm. However, when strong theta reset was considered, the phase at which stimulation was delivered did not influence the outcome. In the presence of an even weaker theta input, mimicking a pathological state that completely abolishes theta-gamma oscillations, only pulse train stimulation could restore physiological theta-gamma oscillations during the stimulation period. As for the previous results, this effect was phase-dependent only when theta reset was not included.

These results provide a new framework to interpret neurostimulation interventions that interfere with hippocampal oscillations and aim at improving memory function. It can be further extended to investigate the effects of more complex neurostimulation protocols, and the impact of stimulation location and amplitude on the observed network dynamics.

## Results

### A computational model of the hippocampal formation with dynamical theta input

We first developed a computational model of hippocampal circuits, able to generate both theta-nested gamma oscillations and theta phase reset. To achieve this, we combined an existing conductance-based model of the hippocampal formation (*Aussel et al., 2018*) with a set of Kuramoto phase oscillators (*Kuramoto, 1984*) that were used to model the dynamical theta input originating from pacemaker neurons in the medial septum (*Wang, 2002*, *Figure 1A*). Such ensembles of oscillators are designed to exhibit a strong phase reset of their collective rhythm in response to a perturbation (*Levnajić and Pikovsky, 2010*). The hippocampal model contained excitatory and inhibitory neuronal populations in the entorhinal cortex (EC), dentate gyrus (DG), CA3 and CA1 fields within a coronal slice of the human hippocampus (*Figure 1B–C*). The output of the hippocampal formation (i.e. the activity of CA1 pyramidal neurons) was provided as an input to the Kuramoto oscillators, simulating the hippocampal-septal projections through the fornix (*Williams and Givens, 2003*; *Nuñez and Buño, 2021*; *Takeuchi et al., 2021*). The oscillations produced by the collective behavior of Kuramoto oscillators represented a population average of their activity: highly synchronized and desynchronized states generated respectively high and low amplitude theta oscillations (*Figure 1D*). The number of oscillators was chosen so that their synchronization level (i.e. their order parameter) and their frequency distribution were sufficiently close to their asymptotic behavior for a large number of oscillators (*Figure 1—figure supplement 1*).

Biologically, GABAergic neurons from the medial septum project to the EC, CA3, and CA1 fields of the hippocampus (*Tóth et al., 1993*; *Hajós et al., 2004*; *Manseau et al., 2008*; *Hangya et al., 2009*;

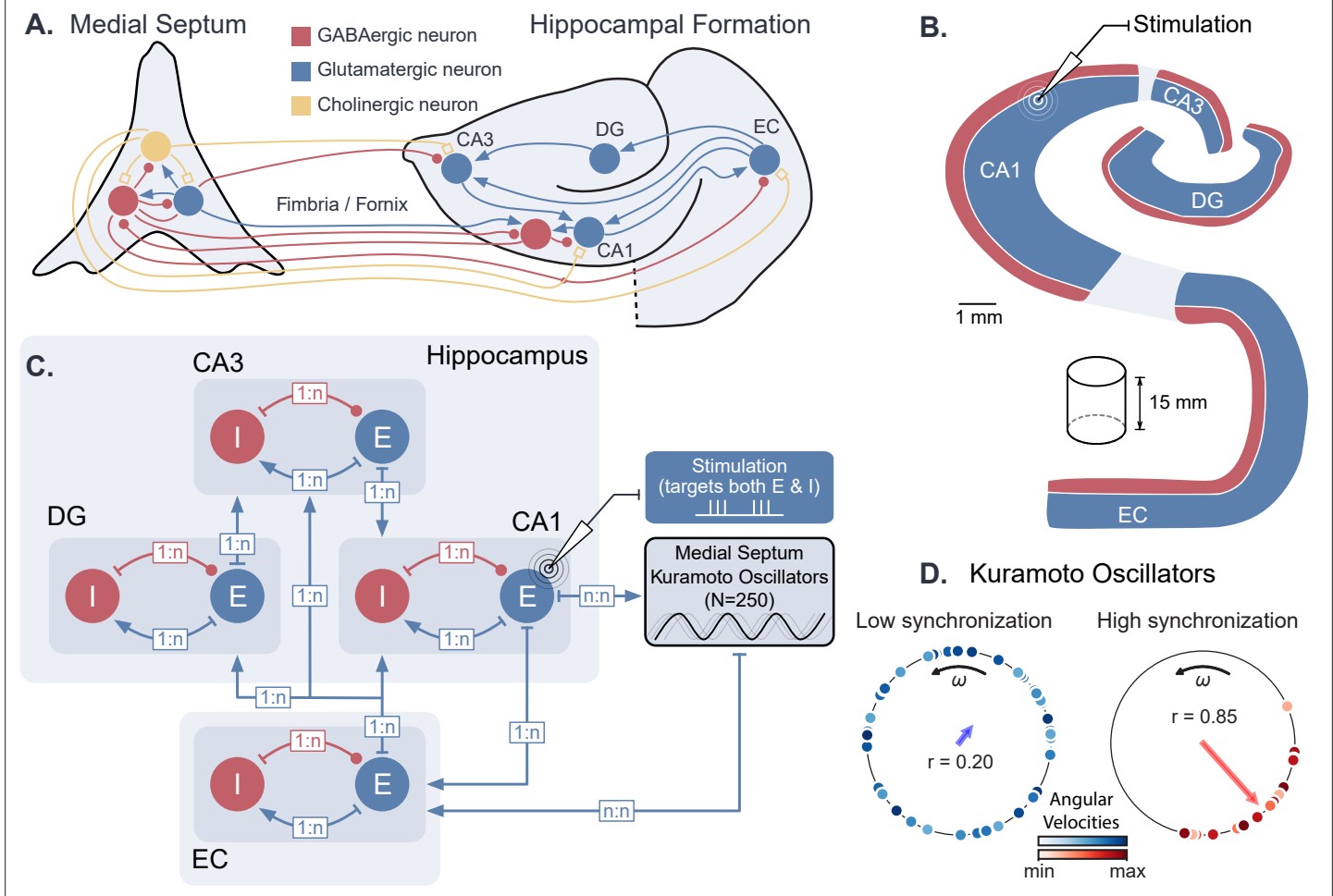

**Figure 1.** Dynamical computational model of the hippocampal formation and medial septum oscillatory drive. (**A**) Anatomical representation of the neuronal types and interconnections within and between the medial septum and the hippocampal formation (EC: entorhinal cortex, DG: dentate gyrus, CA3 and CA1 fields of the hippocampus). (**B**) Simplified anatomy of the hippocampal formation, modeled as a 15-mm-thick cylindrical slice, with spatially segregated excitatory and inhibitory neurons (blue: excitatory neurons, consisting of granule cells in DG and pyramidal cells in other areas; red: inhibitory basket cells). (**C**) Model architecture and connectivity. Each area is comprised of one excitatory and one inhibitory neuronal population. Theta drive is provided through input from the medial septum, which is modeled as a set of 250 Kuramoto oscillators and receives feedback connections from CA1. Electrical stimulation Is modeled as an intracellular current affecting both excitatory and inhibitory populations in the targeted area (shown here for CA1). (**D**) Illustration of Kuramoto oscillators with two different levels of synchronization. Each dot represents one oscillator, its position on the circle indicates its phase and its color its angular velocity. Higher synchronization corresponds to a clustering of the dots around a similar phase. '*r*' indicates the order parameter, which is a measure of synchronization.

The online version of this article includes the following figure supplement(s) for figure 1:

**Figure supplement 1.** Influence of the number of Kuramoto oscillators on their collective dynamics.

*Unal et al., 2015*; *Müller and Remy, 2018*). Although the respective roles of these different projections are not fully understood, previous computational studies have suggested that the direct projection from the medial septum to CA1 is not essential for the production of theta in CA1 microcircuits (*Mysin et al., 2019*). Since our modeling of the medial septum is only used to generate a dynamic theta rhythm, we opted for a simplified representation where the medial septum projects only to the EC, which in turn drives the different subfields of the hippocampus. In our model, Kuramoto oscillators are therefore connected to the EC neurons and they receive projections from CA1 neurons (see Materials and methods for more details).

From a conceptual point of view, our model is thus composed of excitatory-inhibitory (E-I) circuits connected in series, with a feedback loop going through a population of coupled phase oscillators. In the next sections, we first describe the generation of gamma oscillations by individual E-I circuits

(*Figure 2*), and illustrate their behavior when driven by an oscillatory input such as theta oscillations (*Figure 3*). We then present a thorough characterization of the effects of theta input and stimulation amplitude on theta-nested gamma oscillations (*Figure 4* and *Figure 5*). Finally, we present some results on the effects of neurostimulation protocols for restoring theta-nested gamma oscillations in pathological states (*Figure 6* and *Figure 7*).

## Generation of gamma oscillations by E-I circuits

It is well established that a network of interconnected pyramidal neurons and interneurons can give rise to oscillations in the gamma range, a mechanism termed pyramidal-interneuronal network gamma (PING) (*Traub et al., 2004*; *Onslow et al., 2014*; *Segneri et al., 2020*). This mechanism has been observed in several optogenetic studies with gradually increasing light intensity (i.e. under a ramp input) affecting multiple different circuits, such as layer 2–3 pyramidal neurons of the mouse somatosensory cortex (*Adesnik and Scanziani, 2010*), the CA3 field of the hippocampus in rat in vitro slices (*Akam et al., 2012*), and in the non-human primate motor cortex (*Lu et al., 2015*). In all cases, gamma oscillations emerged above a certain threshold in terms of photostimulation intensity, and the frequency of these oscillations was either stable or slightly increased when increasing the intensity further. We sought to replicate these findings with our elementary E-I circuits composed of single-compartment conductance-based neurons driven by a ramping input current (*Figure 2* and *Figure 2—figure supplement 1*). As an example, all the results in this section will be shown for an E-I circuit that has similar connectivity parameters as the CA1 field of the hippocampus in our complete model (see section 'Hippocampal formation: inputs and connectivity' in the Materials and methods).

For low input currents provided to both neuronal populations, only the highly excitable interneurons were activated (*Figure 2A*). For a sufficiently high input current (i.e. a strong input that could overcome the inhibition from the fast-spiking interneurons), the pyramidal neurons started spiking as well. As the amplitude of the input increased, the activity of both neuronal populations became synchronized in the gamma range, asymptotically reaching a frequency of about 60 Hz (*Figure 2A* bottom panel). Decoupling the populations led to the abolition of gamma oscillations (*Figure 2B*), as neuronal activity was determined solely by the intrinsic properties of each cell. Interestingly, when the ramp input was provided solely to the excitatory population, we observed that the activity of the pyramidal neurons preceded the activity of the inhibitory neurons, while still preserving the emergence of gamma oscillations (*Figure 2—figure supplement 1A*). As expected, decoupling the populations also abolished gamma oscillations, with the excitatory neurons spiking a frequency determined by their intrinsic properties and the inhibitory population remaining silent (*Figure 2—figure supplement 1B*).

To further characterize the intrinsic properties of individual inhibitory and excitatory neurons, we derived their input-frequency (I-F) curves, which represent the firing rate of individual neurons in response to a tonic input (*Figure 2—figure supplement 2A*). We observed that for certain input amplitudes, the firing rates of both types of neurons was within the gamma range. Interestingly, in the absence of noise, each population could generate by itself gamma oscillations that were purely driven by the input and determined by the intrinsic properties of the neurons (*Figure 2—figure supplement 2B*). Adding stochastic Gaussian noise in the membrane potential disrupted these artificial oscillations in decoupled populations (*Figure 2—figure supplement 2C*). All subsequent simulations were run with similar noise levels to prevent the emergence of artificial gamma oscillations.

Another potent way to induce gamma oscillations is to drive fast-spiking inhibitory neurons using pulsed optogenetic stimulation at gamma frequencies, a strategy that has been used both in the neocortex (*Cardin et al., 2009*) and hippocampal CA1 (*Iaccarino et al., 2016*). In particular, Cardin and colleagues systematically investigated the effect of driving either excitatory or fast-spiking inhibitory neocortical neurons at frequencies between 10 and 200 Hz (*Cardin et al., 2009*). They showed that fast-spiking interneurons are preferentially entrained around 40–50 Hz, while excitatory neurons respond better to lower frequencies. To verify the behavior of our model against these experimental data, we simulated pulsed optogenetic stimulation as an intracellular current provided to our reduced model of a single E-I circuit. Stimulation was applied at frequencies between 10 and 200 Hz to excitatory cells only, to inhibitory cells only, or to both at the same time (*Figure 2—figure supplement 3*). The population firing rates were used as a proxy for the local field potentials (LFP), and we computed the relative power in a 10 Hz band centered around the stimulation frequency, similarly to the method proposed in *Cardin et al., 2009*. When presented with continuous stimulation across a range of

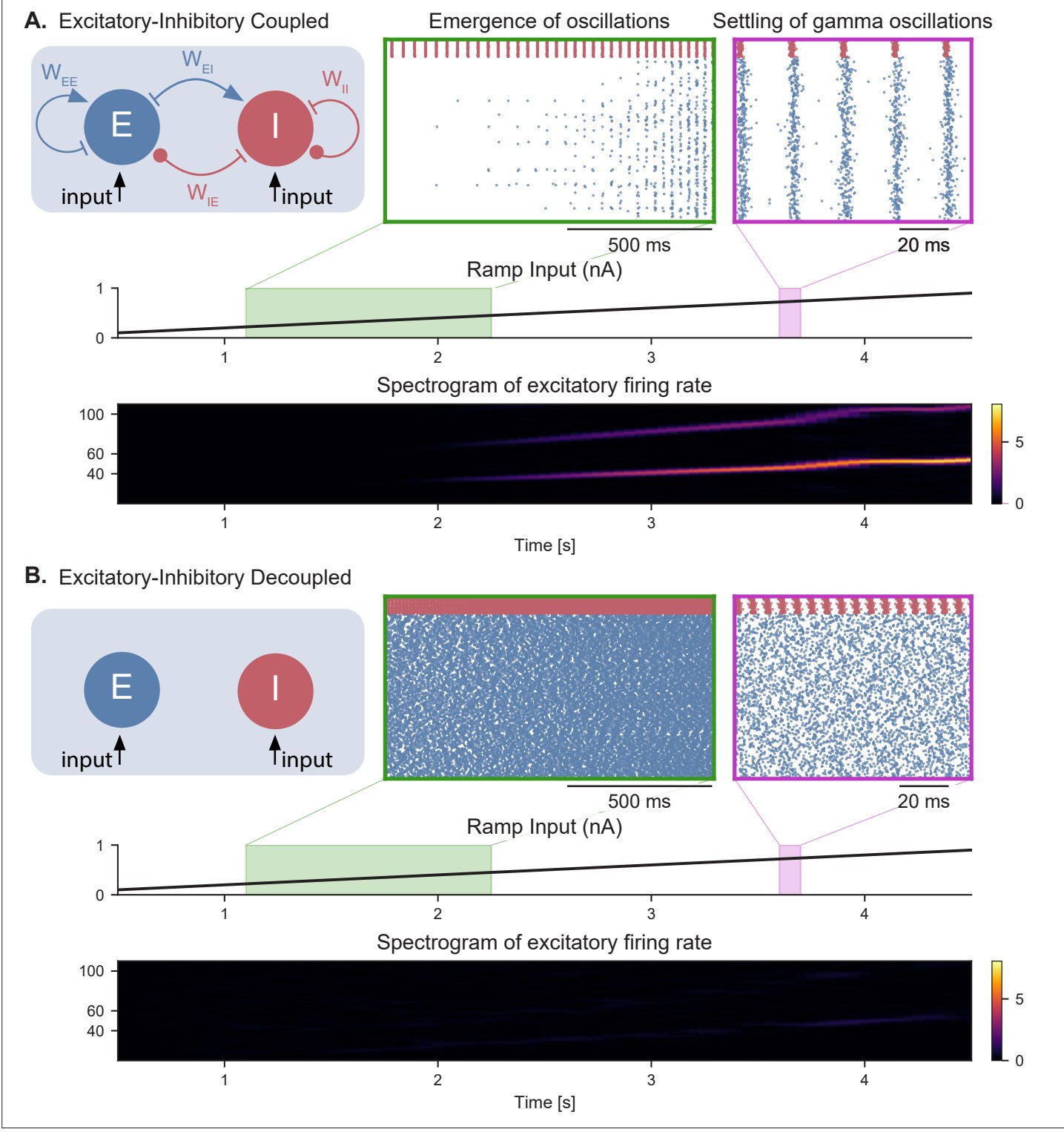

**Figure 2.** Emergence of gamma oscillations in coupled excitatory-inhibitory populations under ramping input to both populations. (**A**) Two coupled populations of excitatory pyramidal neurons ($N_E$ = 1000) and inhibitory interneurons ($N_I$ = 100) are driven by a ramping current input (0 nA to 1 nA) for 5 s. As the input becomes stronger, oscillations start to emerge (shaded green area), driven by the interactions between excitatory and inhibitory populations. The green inset shows the raster plot (neuronal spikes across time) of the two populations during the green shaded period (red for inhibitory; blue for excitatory). When the input becomes sufficiently strong (shaded magenta area), the populations become highly synchronized and produce oscillations in the gamma range (at approximately 50 Hz). The spectrogram (bottom panel) shows the power of the instantaneous firing rate of the pyramidal population as a function of time and frequency. It reveals the presence of gamma oscillations that emerge around 2 s and increase

*Figure 2 continued on next page*

*Figure 2 continued*

in frequency until 4 s, when they settle at approximately 60 Hz. (**B**) Similar depiction as in panel A. with the pyramidal-interneuronal populations decoupled. The absence of coupling leads to the abolition of gamma oscillations, each cell spiking activity being driven by its own inputs and intrinsic properties.

The online version of this article includes the following figure supplement(s) for figure 2:

**Figure supplement 1.** Emergence of gamma oscillations in coupled excitatory-inhibitory populations under ramping input to the excitatory population.

**Figure supplement 2.** Cell-intrinsic spiking activity in decoupled excitatory and inhibitory populations under ramping input.

**Figure supplement 3.** Neuronal entrainment of fast-spiking interneurons by pulsed optogenetic stimulation in the high gamma range.

frequencies in the gamma range, interneurons showed the greatest degree of gamma power modulation (*Figure 2—figure supplement 3*). Furthermore, when the stimulation was delivered to the excitatory population, the relative power around the stimulation frequency dropped significantly in frequencies above 10 Hz, similar to the reported experimental data (*Cardin et al., 2009*). The main difference between our simulation results and these experimental data is the specific frequencies at which fast-spiking interneurons showed resonance, which was around 40 Hz in the mouse barrel cortex and around 90 Hz in our model, a fast gamma rhythm. This could be attributed to several factors, such as differences in the cellular properties between cortical and hippocampal fast-spiking interneurons, or the differences between the size of the populations and their relevant connectivity in the cortex and the hippocampus.

## Theta-gamma oscillations and theta phase reset under dynamical theta input

Once we validated that our elementary E-I circuits were able to generate gamma oscillations under an input of sufficiently high amplitude, we studied the behavior of the whole model when the theta input was dynamically provided by the Kuramoto oscillators as described above (*Figure 1*). As in the original model (*Aussel et al., 2018*), the input theta rhythm drove the network to produce spiking activity in the gamma range preferentially around the peak of theta in all excitatory and inhibitory populations (*Figure 3A* and *Figure 3—figure supplement 1*). Spectrograms of the CA1 population firing rates revealed that these bursts of activity around each theta peak were characterized by oscillations around 60 Hz (*Figure 3A*). This was confirmed by power spectral densities, which showed a clear peak at 4 Hz (corresponding to the theta drive) and increases in gamma band activity between 40 and 80 Hz (*Figure 3B*). To quantify PAC between theta and gamma oscillations, we used the modulation index (MI) (*Tort et al., 2008*; *Tort et al., 2010*), which has been shown to outperform other similar measures (*Hülsemann et al., 2019*). However, we discovered that this metric would give erroneous results in our simulated datasets due to the absence of certain frequency components. To overcome this limitation and avoid artifacts, uniform noise was added to the firing rates prior to computing the MI (*Figure 3—figure supplement 2*). We first visualized the quantification of PAC using the comodulogram, which indicates the MI as a function of two frequencies corresponding to the modulating phase signal and the modulated amplitude signal. This analysis confirmed that gamma-band signals between 45 and 75 Hz were modulated by lower theta frequencies between 3 and 6 Hz, which we quantified by computing a global MI in frequency ranges encompassing theta (3–9 Hz) and gamma (40–80 Hz) (*Figure 3—figure supplement 2B*). To refine this analysis, we also computed a similar MI between the amplitude of various gamma frequencies and the phase of theta, which indicated that oscillations between 40 and 80 Hz occur preferentially around the peak of theta (*Figure 3C* and *Figure 3—figure supplement 2B*).

Next, we verified that our model was able to display theta phase reset during single-pulse stimulation (depolarizing pulse, 1 ms duration). This mechanism is tightly linked with the concept of PRC, which characterizes the phase delay or advancement that follows a single pulse delivered to an oscillatory system, as a function of the phase at which this input is delivered. Although there is no direct measurement of the PRC of septal neurons, such characterizations have been performed for individual pyramidal cells in the CA3 and CA1 fields of the hippocampus (*Lengyel et al., 2005*; *Kwag and Paulsen, 2009*; *Akam et al., 2012*). These PRCs appear biphasic and show a phase advancement (respectively delay) for stimuli delivered in the ascending (respectively descending) slope of theta. We modeled this behavior by a specific term (which we called the phase response function) in the general

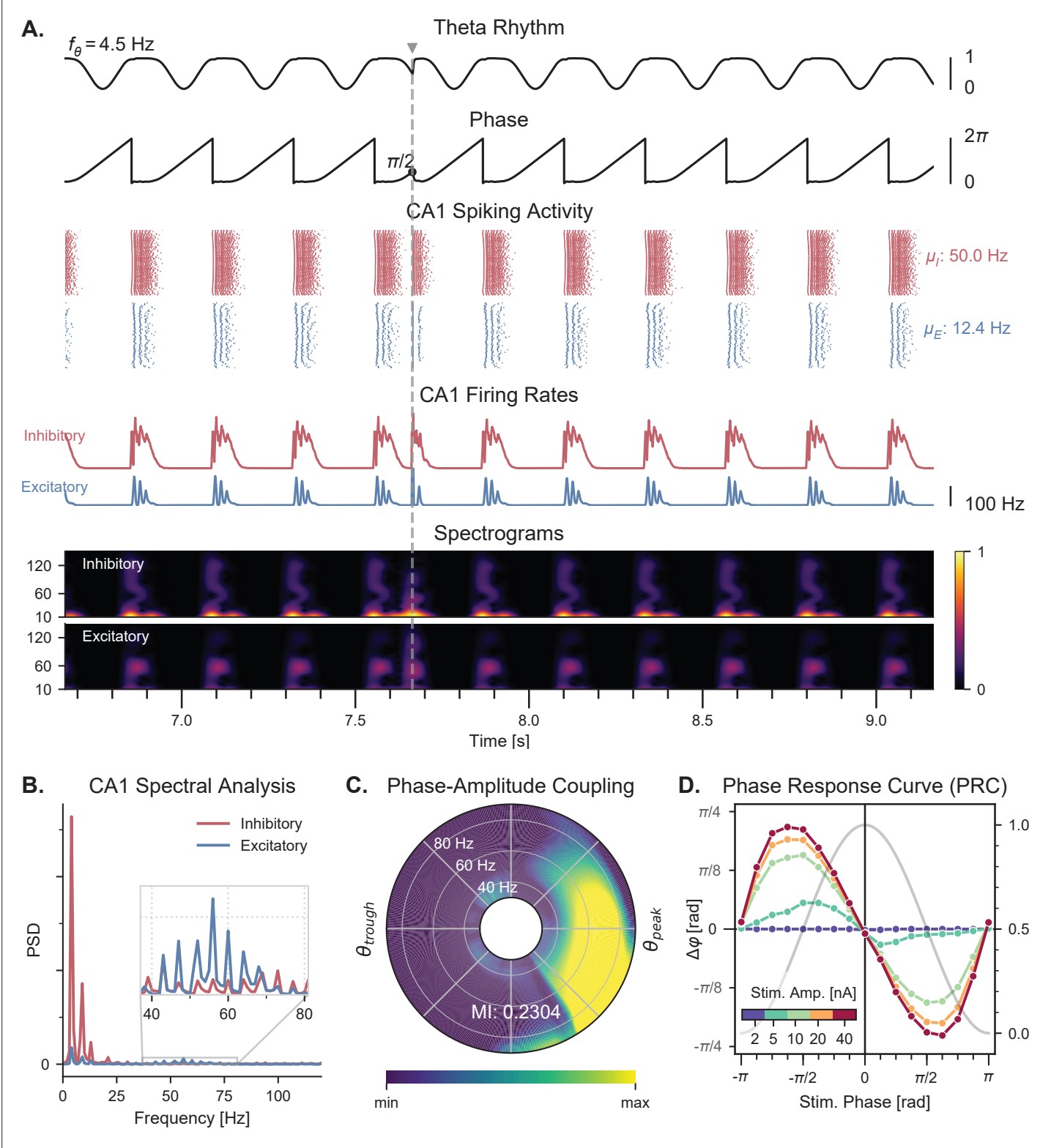

**Figure 3.** Theta-nested gamma oscillations and theta phase reset in response to a single stimulation pulse. (**A**) Representative example of the network behavior, spontaneously producing theta-nested gamma oscillations and characterized by a reset of the theta phase following a single stimulation pulse (vertical grey line, applied here in CA1 at a theta phase of $\pi/2$, that is in the middle of the descending slope). Top to bottom: theta rhythm originating from the medial septum and provided as an input to the EC ($f_\theta$: mean oscillation frequency), instantaneous phase of the theta rhythm, raster plots indicating the spiking activity of CA1 excitatory (blue) and inhibitory (red) neurons ($\mu_E$ and $\mu_I$: mean firing rates within the shaded area), average

*Figure 3 continued on next page*

*Figure 3 continued*

population firing rates in CA1 (computed as a windowed moving average with a sliding window of 100ms with 99% overlap), spectrograms for each CA1 population (windowed short-time fast Fourier transform using a Hann sliding window: 100ms with 99% overlap). Spectrograms show gamma oscillations (around 60 Hz) modulated by the underlying theta rhythm (~4 Hz), indicating theta-gamma PAC. Theta phase reset after stimulation is associated with a rebound of spiking activity and theta-nested gamma oscillations. (**B**) Power spectral densities of the CA1 firing rates. Theta peaks are found at 4 Hz for excitatory and inhibitory cells. Gamma activity is located between 40 and 80 Hz. (**C**) PAC as a function of theta phase and gamma frequency. The polar plot represents the amplitude of gamma oscillations (averaged across all theta cycles, see Materials and methods) at each phase of theta (theta range: 3–9 Hz, phase indicated as angular coordinate) and for different gamma frequencies (radial coordinate, binned in 10 Hz ranges), indicating that gamma oscillations between 40 and 80 Hz occur preferentially around the peak of theta. The MI gives an overall quantification of how the phase of low-frequency oscillations (3–9 Hz) modulates the amplitude of higher-frequency oscillations (40–80 Hz) (see Materials and methods and *Figure 3— figure supplement 2*). (**D**) PRC in response to a single stimulation pulse applied in CA1 at various phases of the ongoing theta rhythm and for various stimulation amplitudes (color-coded). The phase difference (left y-axis) shows the theta phase induced by the stimulation pulse (computed 2.5ms after the pulse), compared to the phase computed at the same time in a scenario without stimulation. Positive and negative phase differences indicate phase advances and delays, respectively. The grey trace shows the normalized amplitude of theta (right *y* axis) for different phases, used to indicate the peak and trough of the rhythm. Stimulation applied in the ascending slope of theta ($[-\pi, 0]$) produced a phase advance and accelerated the rhythm towards its peak (0 radians). Conversely, stimulation during the descending slope ($[0, \pi]$) produced a phase delay that slowed down the rhythm. Higher stimulation amplitudes yielded a stronger effect.

The online version of this article includes the following figure supplement(s) for figure 3:

**Figure supplement 1.** Whole-network behavior producing theta-nested gamma oscillations and theta phase reset in response to a single stimulation pulse.

**Figure supplement 2.** Quantification of PAC with and without noise.

**Figure supplement 3.** Network behavior generated by Kuramoto oscillators with non-physiological phase response functions.

**Figure supplement 4.** A neural mass model of coupled excitatory and inhibitory neurons driven by Kuramoto oscillators generates theta-nested gamma oscillations and theta phase reset.

equation of the Kuramoto oscillators (see methods, *Equation 1*). Importantly, introducing a phase offset in the phase response function disrupted theta-nested gamma oscillations (*Figure 3—figure supplement 3*), which suggests that the septohippocampal circuitry must be critically tuned to be able to generate such oscillations. The strength of phase reset could also be adjusted by a gain that was manually tuned. In the presence of the physiological phase response function and of a sufficiently high reset gain, a single stimulation pulse delivered to all excitatory and inhibitory CA1 neurons could reset the phase of theta to a value close to its peaks (*Figure 3A*). We computed the PRC of our simulated data for different stimulation amplitudes and validated that our neuronal network behaved according to the phase response function set in our Kuramoto oscillators (*Figure 3D*). It should be noted that including this phase reset mechanism affected the generated theta rhythm even in the absence of stimulation, extending the duration of the theta peak and thereby slowing down the frequency of the generated theta rhythm.

Importantly, our approach is generalizable and can be applied to other models producing theta-nested gamma oscillations. For instance, we adapted the neural mass model by Onslow and colleagues (*Onslow et al., 2014*), replaced the fixed theta input by a set of Kuramoto oscillators, and demonstrated that it could also generate theta phase reset in response to single-pulse stimulation (*Figure 3—figure supplement 4*). These results illustrate that the general behavior of our model is not specific to the tuning of individual parameters in the conductance-based neurons, but follows general rules that are captured by the level of abstraction of the Kuramoto formalism.

Overall, we successfully developed a new model of the hippocampal formation able to exhibit both theta-nested gamma oscillations and theta phase reset in response to stimulation. We then decided to explore further the effects of various stimulation protocols on its dynamics.

## Effects of theta input and stimulation amplitudes on theta-gamma oscillations

We investigated the behavior of the model across multiple states of varying septal theta input amplitude in response to single-pulse stimulation delivered to CA1. In the absence of theta drive, a single stimulation pulse elicited either zero or two bursts of spiking activity (depending on stimulation amplitude) separated by about 200 ms (*Figure 4A and C*). We first sought to understand the origin of this second burst, as it showed that even a single pulse could induce transient periodic activity around

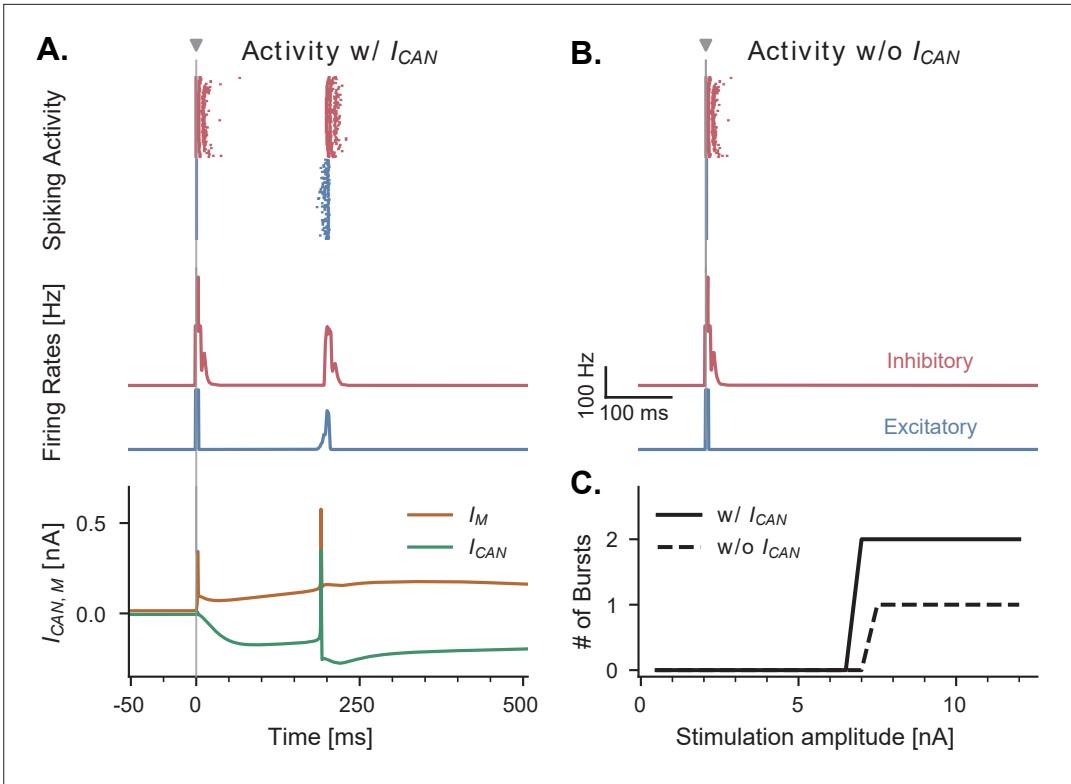

**Figure 4.** CAN and adaptation (M) currents modulate neuronal responses to single-pulse stimulation in the absence of theta input. (**A**) Network response to single-pulse stimulation in the absence of medial septum input. Stimulation (grey vertical line, 10 nA, applied in CA1) induced an instantaneous burst of activity lasting about 20ms in both excitatory and inhibitory CA1 neurons, followed by a secondary burst approximately 200 ms later (raster plot and firing rate traces), associated with specific CAN- and M- currents dynamics (bottom traces, illustrated for a representative CA1 excitatory neuron). Positive ($I_M$) and negative ($I_{CAN}$) currents indicate respectively a hyperpolarizing and depolarizing effect on the cell membrane potential. (**B**) Similar representation as in A., but in the absence of CAN channel. Stimulation induced only a single burst of activity, indicating that CAN channels are necessary to observe a rebound of activity. (**C**) Number of bursts in CA1 spiking activity following a single stimulation pulse at various amplitudes (x-axis), shown both in the presence and absence of CAN channels in excitatory neurons. The absence of the CAN current leads to the abolition of the second burst, irrespective of stimulation amplitude.

5 Hz in our model, a frequency within the theta range. A previous model has shown that the presence of Calcium-Activated Non-specific cationic (CAN) currents can lead to self-sustained theta oscillations in the hippocampus (*Giovannini et al., 2017*). Moreover, this study showed a direct link between the increased excitation provided by the CAN current and the spike-frequency adaptation properties of the M current, directly affecting transitions from an asynchronous low-firing regime to synchronous bursting. We tested the role of the CAN current in the response to single-pulse stimulation by completely removing it from our simulations, which abolished the second burst of activity (*Figure 4B*). Moreover, the time interval separating the two bursts likely resulted from the interplay between the depolarizing CAN current and hyperpolarizing M current (*Figure 4A*).

In the presence of dynamic theta input, the effects of single-pulse stimulation depended both on theta input amplitude and stimulation amplitude, highlighting different regimes of network activity (*Figure 5* and *Figure 5—figure supplement 1*, *Figure 5—figure supplement 2*, *Figure 5—figure supplement 3*). For low theta input, theta-nested gamma oscillations were initially absent and could not be induced by stimulation (*Figure 5A*). At most, the stimulation could only elicit a few bursts of spiking activity that faded away after approximately 250ms, similar to the rebound of activity seen in the absence of theta drive. For increasing theta input, the network switched to an intermediate regime: upon initialization at a state with no spiking activity, it could be kicked to a state with self-sustained theta-nested gamma oscillations by a single stimulation pulse of sufficiently high amplitude

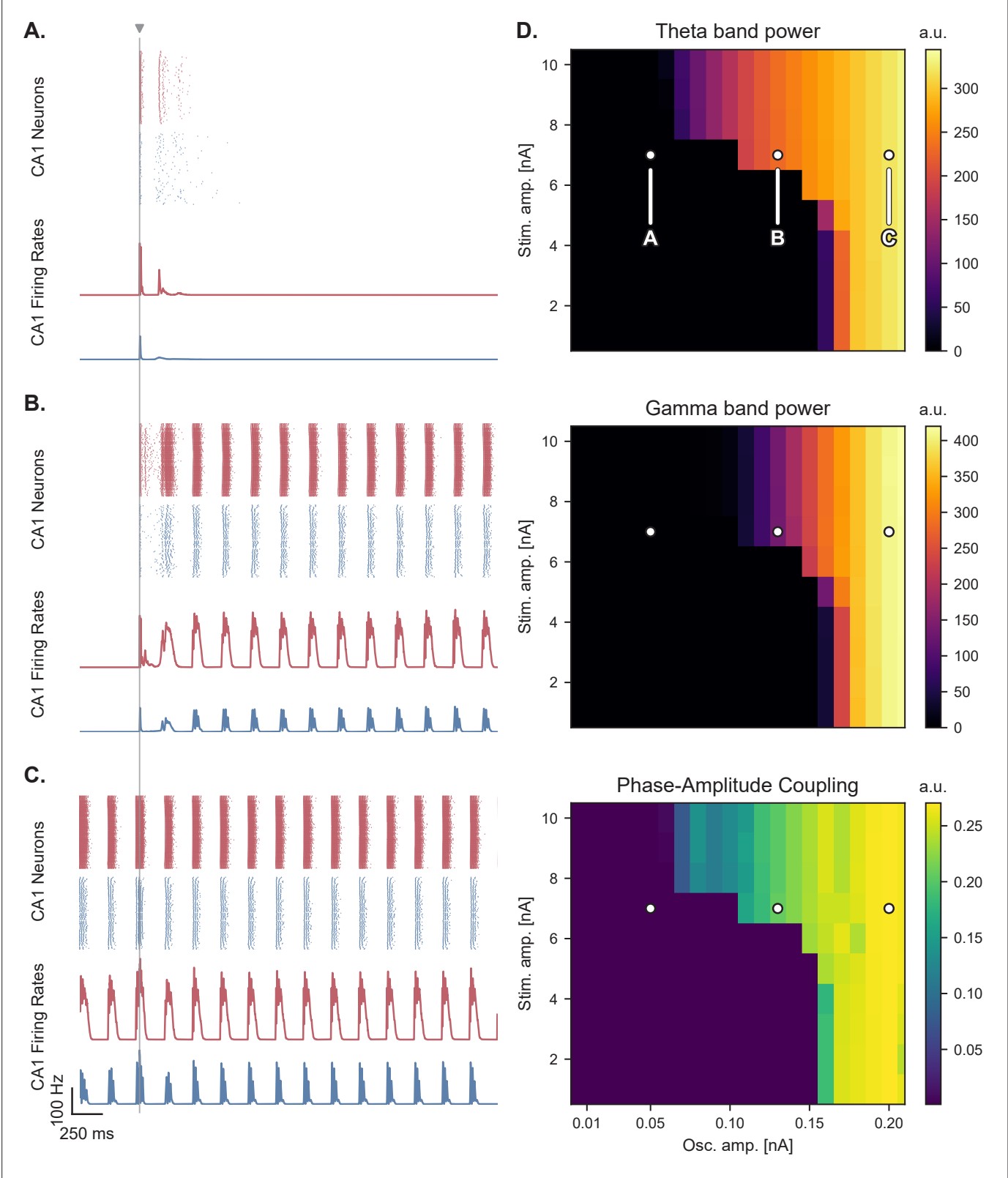

**Figure 5.** Medial septum oscillatory drive and stimulation amplitude govern the steady-state response to single-pulse stimulation. (**A-C**) Network responses to single-pulse stimulation (vertical line) under medial septum input (with phase reset), shown for different amplitudes of the medial septum oscillatory drive (A-C: increasing oscillator amplitudes). (**A**) Low oscillatory input: stimulation induces only two bursts of spiking activity as in *Figure 4*. (**B**) Medium oscillatory input: a single stimulation pulse switches network behavior from no activity to sustained oscillations driven by the medial septum.

*Figure 5 continued on next page*

*Figure 5 continued*

(**C**) Higher oscillatory input: theta drive is capable of inducing self-sustained theta-nested gamma oscillations. In this case, stimulation is delivered at the peak of theta oscillations and does not show a pronounced effect on theta-gamma oscillations. (**D**) Steady-state response to single-pulse stimulation as a function of medial septum oscillatory input (x-axis) and stimulation amplitude (y-axis), characterized by three metrics: theta power (3–9 Hz), gamma power (40–80 Hz), and PAC (quantified using the MI). White dots: parameter combinations corresponding to panels A-C.

The online version of this article includes the following figure supplement(s) for figure 5:

**Figure supplement 1.** Whole-network behavior in response to single-pulse stimulation under low theta oscillatory drive.

**Figure supplement 2.** Whole-network behavior in response to single-pulse stimulation under medium theta oscillatory drive.

**Figure supplement 3.** Whole-network behavior in response to single-pulse stimulation under higher theta oscillatory drive.

**Figure supplement 4.** CAN currents are necessary for the production of self-sustained theta-gamma oscillations in response to single-pulse stimulation.

**Figure supplement 5.** Synchronization level of the oscillators minimally affects the production of theta-nested gamma oscillations.

(*Figure 5B*). This regime existed for a range of septal theta inputs located just below the threshold to induce self-sustained theta-gamma oscillations without additional stimulation, as characterized by the post-stimulation theta power, gamma power, and theta-gamma PAC (*Figure 5D*). Removing CAN currents from all areas of the model abolished this behavior (*Figure 5 - figure supplement 4*), which is interesting given the role of this current in the multistability of EC neurons (*Egorov et al., 2002*; *Fransén et al., 2006*) and in the intrinsic ability of the hippocampus to generate theta-nested gamma oscillations (*Giovannini et al., 2017*). For the highest theta input, the network became able to spontaneously generate theta-nested gamma oscillations, even when initialized at a state with no spiking activity and without additional neurostimulation *Figure 5C*.

## Neurostimulation for restoring theta-gamma oscillations in pathological states

Based on the above analyses, we considered two pathological states: one with a moderate theta input (i.e. moderately weak projections from the medial septum to the EC) that allowed the initiation of self-sustained oscillations by single stimulation pulses (*Figure 5*, point B), and one with a weaker theta input characterized by the complete absence of self-sustained oscillations even following transient stimulation (*Figure 5*, point A). In each case, we sought to assess whether single-pulse or pulse train stimulation could induce or restore theta-nested gamma oscillations and whether this effect depended on the phase at which stimulation was delivered (i.e. at the peak or trough of the theta cycle). We hypothesized that any possible phase relationship would also depend on the phase reset mechanism. To test this hypothesis, we ran a series of simulations using two different models: one without phase reset and one with strong phase reset (i.e. the reset gain was set at the value used in *Figure 3*).

In the case of a moderate theta input and in the presence of phase reset, delivering a pulse at either the peak or trough of theta could induce theta-nested gamma oscillations (*Figure 6A and C*). By contrast, in the absence of phase reset, only stimulation delivered at the peak of theta was able to induce such oscillations, after some time delay (*Figure 6B and D*). Quantification of these results in terms of theta power, gamma power, and theta-gamma PAC showed strongly similar responses between stimulation delivered at the peak or trough of theta with phase reset enabled, similar but weaker responses with stimulation delivered at the peak of theta with phase reset disabled, and no response in the case of trough stimulation with phase reset disabled (*Figure 6E*).

In the case of a weak theta input that completely abolished neuronal oscillations, we delivered stimulation pulses continuously at a frequency matching that of the underlying theta rhythm for a duration of 2 s, in order to restore physiological oscillations (*Figure 7*). The stimulation onset was timed to either the peak or trough of the ongoing theta cycle. The continuous delivery of pulses produced similar results as single-pulse stimulation. With phase reset, pulse train stimulation restored theta-nested gamma oscillations within the whole network, irrespective of the phase of stimulation onset (*Figure 7A and C*). Notably, stimulation delivered at the trough forced a reset of the phase of theta rhythm and led to the subsequent delivery of pulses at the peak of theta. Interestingly, in the absence of phase reset, peak-targeted stimulation induced theta-gamma oscillations in all network areas easily, while trough-targeted stimulation created artificial bursts that propagated in other areas with difficulty, requiring multiple pulses to achieve a fraction of the results of peak-targeted stimulation

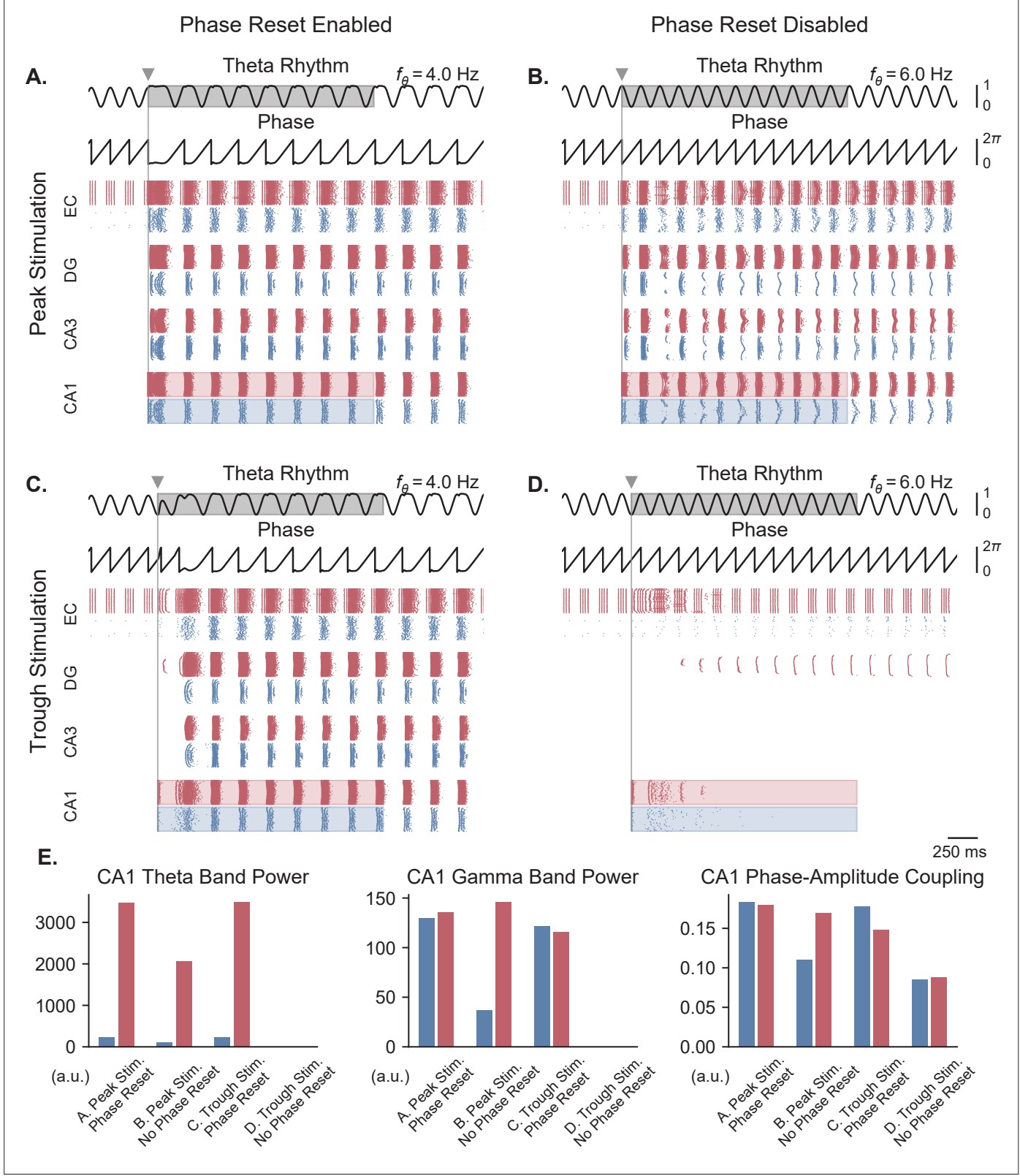

**Figure 6.** Single-pulse stimulation phase differentially affects network responses depending on the presence of theta phase reset. All results shown here were obtained for parameters from *Figure 5B* (theta oscillation amplitude: 0.13 nA, stimulation amplitude: 7.0 nA). A single stimulation pulse was delivered at the peak (**A, B**) or trough (**C, D**) of the underlying theta rhythm, either in the presence (**A, C**) or absence (**B, D**) of theta phase reset. With phase reset, both peak and trough stimulation switch network behavior from no activity to sustained oscillations. Without phase reset, only peak

*Figure 6 continued on next page*

*Figure 6 continued*

stimulation can induce sustained oscillations. (**E**). Quantification of theta power, gamma power, and PAC (measured using the MI) in CA1 excitatory (blue) and inhibitory (red) populations in all four cases (metrics are computed in the shaded areas of panels A-D).

(*Figure 7B and D*). Comparing these simulations based on theta power, gamma power, and theta-gamma PAC within CA1 (*Figure 7E*) showed similar albeit less striking differences as single-pulse stimulation, possibly because CA1 was driven directly by the stimulation. Notably, gamma power in the absence of theta phase reset, was higher when utilizing pulse trains. These differences were even more pronounced in areas other than CA1, as can be seen by the gradual emergence of oscillations for trough stimulation without phase reset (*Figure 7D*).

## Discussion
### Highlights

In summary, we have developed a novel computational model to investigate the effects of electrical stimulation on a slice of the hippocampal formation, incorporating two important features related to memory: theta-nested gamma oscillations and theta phase reset. The key innovation compared to previous models (e.g. *Aussel et al., 2018*) is the introduction of a set of abstract Kuramoto oscillators, which represent pacemaker neurons in the medial septum and are interfaced with biophysically realistic neuronal models in the hippocampus. From a methodological point of view, this hybrid interfacing between two levels of abstraction represents an innovation in itself and could be applied to other systems or brain structures that are driven by dynamical rhythms. The main outcomes reported here relate to the importance of the theta reset mechanism when examining the effects of neurostimulation on hippocampal oscillations.

A very interesting finding concerns the behavior of the model in response to single-pulse stimulation for certain values of the theta amplitude (*Figure 5*). For low theta amplitudes, a single stimulation pulse was capable of switching the network behavior from a state with no spiking activity to one with prominent theta-nested gamma oscillations. Whether such an effect can be induced in vivo in the context of memory processes remains an open question. Nevertheless, delivering a single stimulation pulse bilaterally to the human hippocampus during a memory task is sufficient to impair memory encoding (*Lacruz et al., 2010*), suggesting that even single-pulse stimulation can indeed have wide network effects that are behaviorally relevant.

The second main finding is that the timing of individual stimulation pulses with respect to the phase of the ongoing theta rhythm matters differently depending on the presence or absence of phase reset (*Figure 6* and *Figure 7*). Human intracranial stimulation data indicate that the receptivity of hippocampal circuits to single-pulse stimulation is modulated by the phase of theta (*Lurie et al., 2022*). A number of studies have also reported conflicting results in terms of memory outcomes, which could potentially be attributed to the induction of phase reset through stimulation (*Suthana et al., 2012*; *Jacobs et al., 2016*). To our knowledge, however, the degree of phase reset that follows each stimulation pulse remains unknown and should be investigated in future experimental studies. From a technological point of view, the two regimes with and without phase reset have opposite predictions concerning the need for closed-loop stimulation protocols that would trigger stimulation in real-time based on the phase of ongoing theta oscillations. Such phase-triggered stimulation would be most useful if the phase reset mechanism remains relatively limited. When this mechanism becomes too strong, regular continuous stimulation appears sufficient to restore physiological theta-gamma oscillations (*Figure 7*).

It should also be noted that the reset gain in our simulations (*Equation 1*, gain $G_{reset}$) was either completely turned off or set at a high value, producing a strong effect that always reset the phase of theta to its peak. In reality, the degree of theta phase reset is dynamic, depending on the environment and associated task requirements (*Rizzuto et al., 2003*; *Mormann et al., 2005*; *Jackson et al., 2008*), and may be affected by neurodegenerative disorders that affect the connections between the hippocampal formation and the medial septum. Importantly, our proposed framework can simulate intermediate values of the reset gain, which should ideally be fitted to experimental data in future applications.

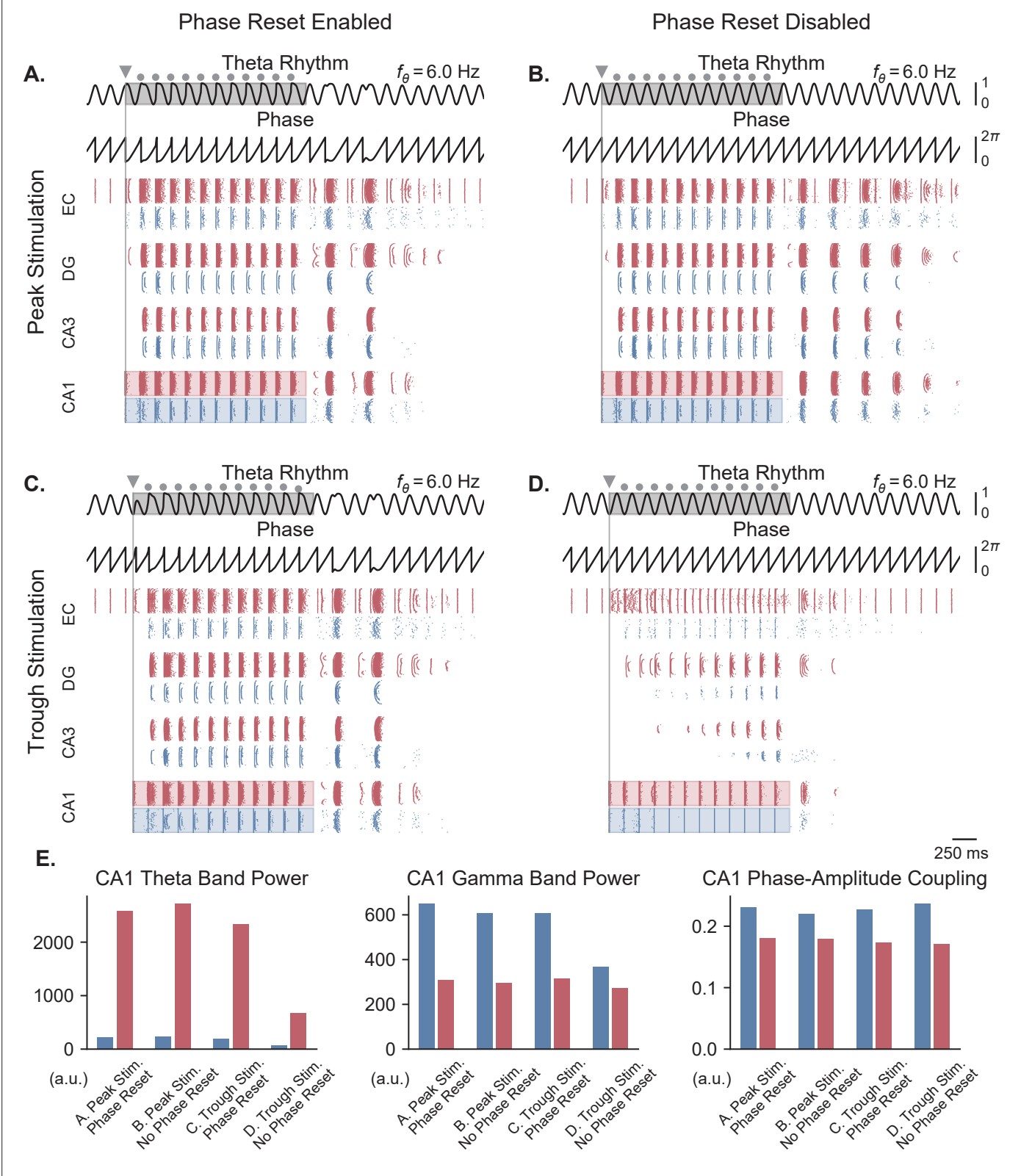

**Figure 7.** Pulse train stimulation restores theta-nested gamma oscillations depending on stimulation timing and theta phase reset. All results shown here were obtained for parameters from *Figure 5A* (theta oscillation amplitude: 0.05 nA; stimulation amplitude: 7.0 nA). Representations are similar to *Figure 6*, with the difference that stimulation consisted of a pulse train delivered at 6 Hz for a duration of 2 s (individual pulses indicated by grey dots, first pulse by a triangle). The pulse train was delivered at the peak (**A, B**) or trough (**C, D**) of the underlying theta rhythm, either in the presence

*Figure 7 continued on next page*

*Figure 7 continued*

(**A, C**) or absence (**B, D**) of theta phase reset. With phase reset, both peak and trough stimulation switch network behavior from no activity to sustained oscillations. Without phase reset, only peak stimulation can induce sustained oscillations. E. Quantification of theta power, gamma power, and PAC (measured using the MI) in CA1 excitatory (blue) and inhibitory (red) populations in all four cases (metrics are computed during the pulse train, within the shaded areas of panels A-D).

Finally, we modeled pathological states by reducing the maximum amplitude of the theta input (*Equation 2*, gain $G_\theta$) until theta-nested gamma oscillations were impaired or even abolished (*Figure 5*). This choice was meant to simulate neurodegeneration in the medial septum, which is known to be affected in Alzheimer's disease, leading to oscillatory disruptions (*Nelson et al., 2014*; *Hampel et al., 2018*; *Takeuchi et al., 2021*). Another possibility would be that neurodegeneration limits the ability of the septal pacemaker neurons to synchronize, thus producing a weaker collective theta rhythm without affecting the maximum amplitude of individual oscillators. Although we simulated this change in our model by reducing the synchronization parameter, the effects on hippocampal oscillations were less pronounced (*Figure 5 - figure supplement 5*). Linking these different modeling parameters to experimental biomarkers will be important in future work.

## Limitations

Even though we took great care in developing a precise representation of the hippocampal formation, the resulting model remains a simplification that could be further enriched. In particular, we deliberately modeled only a single theta generator, while multiple intra- and extra-hippocampal generators are known to co-exist (*Kocsis et al., 1999*; *Hummos and Nair, 2017*). We decided to model septal pacemaker neurons projecting to the EC as the main source of hippocampal theta as reported in multiple experimental studies (*Buzsáki, 2002*; *Buzsáki et al., 2003*; *Hangya et al., 2009*; *Colgin, 2013*). However, experimental findings and previous models have also proposed that direct septal inputs are not essential for theta generation (*Wang, 2002*; *Colgin, 2013*; *Mysin et al., 2019*), but play an important role in phase synchronization of hippocampal neurons. Furthermore, the model does not account for the connections between the lateral and medial septum and the hippocampus (*Takeuchi et al., 2021*). These connections include the inhibitory projections from the lateral to the medial septum and the monosynaptic projections from the hippocampal CA3 field to the lateral septum. An experimental study has highlighted the importance of the lateral septum in regulating the hippocampal theta rhythm *Bender et al., 2015*, an area that has not been included in the model. Specifically, theta-rhythmic optogenetic stimulation of the axonal projections from the lateral septum to the hippocampus was shown to entrain theta oscillations and lead to behavioral changes during exploration in transgenic mice. To account for these discrepancies, our model could be extended by considering more realistic connectivity patterns between the medial/lateral septum and the hippocampal formation, including glutamatergic, cholinergic, and GABAergic reciprocal connections (*Müller and Remy, 2018*), or by considering multiple sets of oscillators each representing one theta generator.

In terms of neuronal cell types, we also made an important simplification by considering only basket cells as the main class of inhibitory interneuron in the whole hippocampal formation. However, it should be noted that many other types of interneurons exist in the hippocampus and have been modeled in various works with higher computational complexity (e.g. *Bezaire et al., 2016*; *Chatzikalymniou et al., 2021*). Among these various interneurons, oriens-lacunosum moleculare (OLM) neurons in the CA1 field have been shown to play a crucial role in synchronizing the activity of pyramidal neurons at gamma frequencies (*Tort et al., 2007*), and in generating theta-gamma PAC (e.g. *Neymotin et al., 2011*; *Ponzi et al., 2023*). Additionally, these cells may contribute to the formation of specific phase relationships within CA1 neuronal populations, through the integration between inputs from the medial septum, the EC, and CA3 (*Mysin et al., 2019*). Future work is needed to include more diverse cell types and detailed morphologies modeled through multiple compartments.

Another limitation of our model concerns synaptic transmission delays, which have been largely neglected and could affect the phase relationships between the medial septum and different hippocampal subfields. Experimental studies have indeed reported time delays in the population activities of connected anatomical structures (i.e. from EC to DG; *Mizuseki et al., 2009*), with pyramidal cells in downstream areas like CA3 and CA1 preferentially firing at different phases of theta (*Dragoi and Buzsáki, 2006*). Propagation effects could also depend on the spatial scale of the model. We also

decided to represent only a thin coronal slice of the hippocampal formation, and it remains unclear how an anatomically accurate model of the whole structure would behave in terms of propagation of spontaneous and electrically-induced neuronal activity.

Importantly, we did not consider learning through synaptic plasticity, even though such mechanisms could drastically modify synaptic conduction for the whole network (*Borges et al., 2017*). Even more interestingly, the inclusion of spike-timing-dependent plasticity would enable the investigation of stimulation protocols aimed at promoting LTP, such as theta-burst stimulation (*Larson and Munkácsy, 2015*). This aspect would be of uttermost importance to make a link with memory encoding and retrieval processes (*Axmacher et al., 2006*; *Tsanov and Manahan-Vaughan, 2009*; *Jutras et al., 2013*) and with neurostimulation studies for memory improvement (*Titiz et al., 2017*; *Solomon et al., 2021*).

From the point of view of neurostimulation, future work is needed to extend the current model to a multi-compartment representation of neurites, since axons are known to be preferentially activated by extracellular electrical stimulation (*Rattay et al., 2003*). Specifically, multi-compartment cable models have been developed to investigate spike initiation and propagation by modeling the axon as a series of resistance-capacitance circuits following its trajectory (*Rattay et al., 2003*; *Joucla and Yvert, 2012*; *Ashida and Nogueira, 2018*). These models are particularly suited to study the effects of extracellular stimulation, which are non-intuitive and depend as a first approximation on the second spatial derivative of the electrical potential along the cell membrane (*Rattay, 1986*; *Rattay et al., 2003*; *McIntyre et al., 2004*; *Rattay et al., 2018*). Here, we have modeled electrical stimulation as an intracellular current applied equally across all neurons in the targeted area, which is extremely simplified but enables computational tractability. Future developments will focus on developing an equivalent multicompartment model with realistic axonal trajectories while making sufficient simplifications to allow for realistic computation times.

Finally, we likened conditions of low theta input to pathological states characteristic of oscillopathies such as Alzheimer's disease, as these conditions disrupted all aspects of theta-gamma oscillations in our model: theta power, gamma power, and theta-gamma PAC (*Figure 5*). However, it should be noted that changes in theta or gamma power in these pathologies are often unclear, and that the most consistent alteration that has been reported in Alzheimer's disease is a reduction of theta-gamma PAC (for review, see *Kitchigina, 2018*). Future work should explore the effects of cellular alterations intrinsic to the hippocampal formation and their impact on theta-gamma oscillations.

## Outlook

Overall, this new model of the hippocampal formation represents a methodologically innovative basis to further explore multiple neurostimulation strategies that target hippocampal oscillations. Moreover, the limitations discussed above represent important avenues for future refinements, which will require significant work to overcome the costs in terms of computational tractability (e.g. modeling the whole hippocampus, or using multicompartment models of axonal trajectories and dendritic trees). Ultimately, such model refinements should allow the investigation of the effects of extracellular stimulation using charge-balanced biphasic pulses and multipolar electrode configurations with realistic electrode geometries. Most importantly, we believe that our current work may also serve as an inspiration for future computational models of oscillopathies (not necessarily limited to the hippocampus), which could benefit from interfacing abstract sets of synchronizing oscillators and investigating their interactions with biophysically-realistic neurons.

## Materials and methods
### Computational model
#### Overall architecture

We aimed to develop a computational model of the hippocampal formation that is able to generate theta-nested gamma oscillations and takes into account the dynamic nature of the theta input from the medial septum, that is the reset of the theta phase following strong activity in the perforant path or fornix (*Buño et al., 1978*; *Williams and Givens, 2003*). To this end, we adapted a previous biophysical model of the human hippocampal formation under fixed sinusoidal input (*Aussel et al., 2018*) and

interconnected it with an abstract representation of the medial septum, modeled as an assembly of Kuramoto oscillators (*Kuramoto, 1984*; *Figure 1*).

More precisely, we modeled a 15-mm-thick coronal slice of the hippocampal formation which comprises the DG, the CA3 and CA1 subfields of the hippocampus, and the EC, all populated with excitatory and inhibitory Hodgkin-Huxley neurons. For the medial septum, we opted for a simplified representation where the medial septum projects only to the EC, which in turn drives the different subfields of the hippocampus (see also 'A computational model of the hippocampal formation with dynamical theta input', second paragraph). In our model, Kuramoto oscillators are therefore connected to the EC neurons and they receive projections from CA1 neurons (see sections below for more details).

## Medial septum: Kuramoto oscillators

The medial septum contains pacemaker neurons (*Varga et al., 2008*; *Hangya et al., 2009*) that synchronize with one another and generate a global theta rhythm. In turn, these pacemaker neurons drive downstream areas that receive projections from the medial septum. Here, we modeled the entire medial septum neuronal assembly as a set of coupled phase oscillators following the Kuramoto model (*Kuramoto, 1984*), which generated the driving theta rhythm that was provided to the hippocampal formation through the EC (*Breakspear et al., 2010*).

Kuramoto oscillators have also been used to investigate the effects of neurostimulation on synchronized brain rhythms in the context of Parkinson's disease or essential tremor (*Tass, 2003*; *Weerasinghe et al., 2019*). Here, each Kuramoto oscillator was described by its phase $\theta_i$, which evolves over time according to the following equation:

$$\frac{d\theta_i}{dt} = \omega_i + \frac{k}{N} \sum_{j=1}^{N} \sin(\theta_j - \theta_i) + G_{reset} X(t) Z(\theta_i) \tag{1}$$

The term $\omega_i$ denotes the natural frequency of oscillator $i$, and is normally distributed around the center frequency $f_0$ and with standard deviation $\sigma$. The synchronization parameter $k$ represents the coupling strength of the group of oscillators, and $N$ is the number of oscillators. Higher values for the synchronization parameter indicate stronger coupling between pairs of oscillators, thus affecting how fast or slow their phases tend to synchronize. The final product $G_{reset} X(t) Z(\theta_i)$ is used to describe the effects of external inputs on the phase of the oscillators. Here, the only external input originates in the projections from CA1 to the medial septum. It is described by the instantaneous firing rate $X(t)$ of the CA1 excitatory population. $G_{reset}$ is an arbitrary gain that determines the strength of the CA1 input to the oscillators. Finally, the function $Z(\theta_i)$ describes how the effects of CA1 inputs on the oscillators depend on the phase of the ongoing theta rhythm.

The coherence and phase of the driving theta rhythm $I_\theta$ were computed using the order parameter $r$ to extract the mean amplitude $A$ and the mean phase $\phi$ of the ensemble of Kuramoto oscillators as follows:

$$\begin{cases} r(t) & = \frac{1}{N} \sum_{i=1}^{N} e^{j\,\theta_i(t)} \\ A(t) & = \mathrm{Re}(r) \\ \phi(t) & = \mathrm{Im}(r) \\ I_\theta(t) & = G_\theta\, A(t)\, \frac{\cos(\phi(t)) + 1}{2} \end{cases} \tag{2}$$

where the mean amplitude $A(t)$ and the mean phase $\phi(t)$ are derived by taking the real and imaginary part of the order parameter respectively, and the output theta rhythm $I_\theta(t)$ is a rectified cosine, multiplied by a gain $G_\theta$ that controls the maximum output amplitude (in nA).

To simulate the effect of the projections from CA1 to the medial septum on the phase of the oscillators, an approximation of the instantaneous firing rate of the CA1 excitatory population was used as the term $X(t)$. To obtain the instantaneous firing rate, we convolved the CA1 population spike train with an exponential kernel using the following equations:

**Table 1.** Full list of the default parameter values for the Kuramoto oscillators.

| Parameter | Value |
| --- | --- |
| Number of oscillators ($N$) | 250 |
| Center frequency ($f_0$) | 6 Hz |
| Standard deviation ($\sigma$) | 0.5 HZ |
| Synchronization ratio ($k/N$) | 15 |
| Phase reset gain ($G_{reset}$) | 4 |
| Peak phase ($\theta_{peak}$) | 0 rad |
| Firing rate time constant ($\tau_{FR}$) | 10 ms |

$$\begin{cases} \dfrac{dX}{dt} &= -\dfrac{X}{\tau_{FR}} \\ X_{t+1} &= X_t + \dfrac{1}{N\,\tau_{FR}}, \forall\, t \in t^S \end{cases} \tag{3}$$

where $t^S$ denotes the ordered set of the spike timings, and $\tau_{FR}$ determines the exponential decay of the kernel, which was set to 10ms to compute population activity (for single neurons, a typical value is 100ms) (*Gerstner et al., 2014*).

Hereafter, we call the term $Z(\theta)$ the phase response function, to distinguish it from the PRC obtained from experimental data or simulations (see section below '*Data Analysis*', '*Phase Response Curve*'). Briefly, the PRC of an oscillatory system indicates the phase delay or advancement that follows a single pulse, as a function of the phase at which this input is delivered. The phase response function $Z(\theta)$ was chosen to mimic as well as possible experimental PRCs reported in the literature (*Lengyel et al., 2005*; *Kwag and Paulsen, 2009*; *Akam et al., 2012*). These PRCs appear biphasic and show a phase advancement (respectively delay) for stimuli delivered in the ascending (respectively descending) slope of theta. To accurately model this behavior, we used the following equation for the phase response function, where $\theta_{peak}$ represents the phase at which the theta rhythm reaches its maximum and the parameter $\phi_{offset}$ controls the desired phase offset from the peak:

$$Z(\theta_i) = -\sin(\theta_i - (\theta_{peak} + \phi_{offset})) \tag{4}$$

An overview of the default parameters for the Kuramoto oscillators and the bidirectional connections between medial septum-hippocampal formation can be found in *Table 1*.

## Hippocampal formation: Hodgkin-Huxley Neurons

The following sections describe in detail how individual neurons and synapses were modeled, and are adapted from the original work by Aussel and colleagues (*Aussel et al., 2018*). Neurons were modeled as conductance-based single compartments, following the Hodgkin-Huxley formalism (*Hodgkin and Huxley, 1952*), in line with previous work (*Aussel et al., 2022*; *Aussel et al., 2018*). The temporal evolution of the membrane potential of each neuron is described by a differential equation whose general form reads:

$$C_m \frac{dV_m}{dt} = -I_L - \sum_{channel} I_{channel} - \sum_{j \in [E,I]} I_{syn_j} + I_\theta + I_{stim} + \eta \tag{5}$$

$I_L$ denotes the leakage current. $I_{channel}$ are currents associated with specific ion channels, namely potassium ($I_K$), fast sodium ($I_{Na}$), and low-threshold calcium ($I_{Ca}$) currents, the CAN current ($I_{CAN}$) (*Giovannini et al., 2017*), and the M-type potassium channel current ($I_M$) responsible for spike adaptation (*Kosenko et al., 2012*; *Sun and Kapur, 2012*; *Kwag et al., 2014*). $I_{syn}$ represents the currents originating from synaptic inputs to the cell and can be either depolarizing (negative sign) or hyperpolarizing (positive sign). $\eta$ is a Gaussian random noise term accounting for other external inputs and synaptic fluctuations. Theta input from the medial septum is modeled as a depolarizing current and is

denoted by $I_\theta$, while electrical stimulation is denoted by $I_{stim}$. Excitatory neurons represent pyramidal cells in EC, CA3, and CA1, and granule cells in DG. They were modeled with $I_{Na}$, $I_K$, and $I_{Ca}$, and

$I_M$ currents, with the addition of $I_{CAN}$ for pyramidal cells. Fast-spiking interneurons in all areas were modeled with $I_{Na}$ and $I_K$ currents. The complete description for all of the above ionic channels and their corresponding currents can be found in (*Giovannini et al., 2017*). Leakage currents followed the following equation:

$$I_L = (g_L \times A) \times (V_m - E_L) \tag{6}$$

where $g_L$ is the maximum leakage conductance, $A$ is the area of the single compartment corresponding to the membrane of a neuron, and $E_L$ is the reversal potential of the leakage channel.

Channel currents $I_K$, $I_M$, $I_{CAN}$ obey the following set of equations:

$$
\begin{aligned}
I_K &= g_K \times A \times n^4 \times (V_m - E_K) \\
I_M &= g_M \times A \times p \times (V_m - E_M) \\
I_{CAN} &= g_{CAN} \times A \times m_{CAN}^2 \times (V_m - E_{CAN})
\end{aligned}
\tag{7}
$$

where $g_K$, $g_M$, $g_{CAN}$ are the maximum conductances for the respective channel, and $n$, $p$, $m_{CAN}$ are the respective gating variables defined by the following differential equations:

$$
\begin{aligned}
\frac{dn}{dt} &= \frac{n_\infty - n}{\tau_n} \\
\frac{dp}{dt} &= \frac{p_\infty - p}{\tau_p} \\
\frac{dm_{CAN}}{dt} &= \frac{m_{CAN,\infty} - m_{CAN}}{\tau_{m_{CAN}}}
\end{aligned}
\tag{8}
$$

For the potassium and CAN currents, the steady-state values for their corresponding gating variables $n_\infty$ and $m_{CAN,\infty}$ and their corresponding time constants $\tau_K$ and $\tau_{CAN}$ depend on the following functions of the transition rate constants:

$$
\begin{aligned}
n_\infty &= \frac{\alpha_n}{\alpha_n + \beta_n} \qquad m_{CAN,\infty} = \frac{\alpha_{m_{CAN}}}{\alpha_{m_{CAN}} + \beta_{m_{CAN}}} \\
\tau_n &= \frac{0.2}{\alpha_n + \beta_n} \qquad \tau_{m_{CAN}} = \frac{0.2}{\alpha_{m_{CAN}} + \beta_{m_{CAN}}}
\end{aligned}
\tag{9}
$$

The sodium current ($I_{Na}$) and calcium current ($I_{Ca}$) follow a similar set of equations:

$$
\begin{aligned}
I_{Na} &= g_{Na} \times A \times m^3 \times h \times (V_m - E_{Na}) \\
I_{Ca} &= g_{Ca} \times A \times m^2 \times h \times (V_m - E_{Ca})
\end{aligned}
\tag{10}
$$

with two gating variables $m$ and $h$ defined by the following differential equations:

$$
\begin{aligned}
\frac{dm}{dt} &= \frac{m_\infty - m}{\tau_m} \qquad \frac{dh}{dt} = \frac{h_\infty - h}{\tau_h} \\
m_\infty &= \frac{\alpha_m}{\alpha_m + \beta_m} \qquad h_\infty = \frac{\alpha_h}{\alpha_h + \beta_h} \\
\tau_m &= \frac{0.2}{\alpha_m + \beta_m} \qquad \tau_h = \frac{0.2}{\alpha_h + \beta_h}
\end{aligned}
\tag{11}
$$

The gating variable of $I_{CAN}$ depends on the calcium concentration within the neuron ($[Ca]_i^{2+}$), given by:

$$
\begin{aligned}
\frac{d[Ca]_i^{2+}}{dt} &= \gamma(I_{Ca}) + \frac{([Ca]_\infty^{2+} - [Ca]_i^{2+})}{\tau_{[Ca]^{2+}}} \\
\gamma(I_{Ca}) &= \frac{-k_u \times I_{Ca}}{2 \times F \times d \times A}
\end{aligned}
\tag{12}
$$

where $\tau_{[Ca]^{2+}}$ = 1s represents the rate of calcium removal from the cell, $[Ca]_\infty^{2+}$ = 0.24 mol/L is the calcium concentration if the calcium channel remains open for a duration of $\Delta T \to \infty$, $k_u = 10^4$ is a unit conversion constant, $F$ is the Faraday constant, and $d$ = 1 μm is the depth at which the calcium is stored inside the cell.

Noise ($\eta$), accounting for random inputs to the network, was simulated as intracellular current acting on the membrane voltage and following the properties of a Gaussian random variable with a mean of 0 µV and a standard deviation of 1000 µV ($\eta_E \sim N(0, 1000)$ µV) for excitatory neurons and a mean of 0 µV and standard deviation of 100 µV ($\eta_I \sim N(0, 100)$ µV) for inhibitory neurons. The ratio of 1:10 between the noise terms was adapted from the original work and it accounts for the higher excitability of the inhibitory neurons as well as the E-I population size ratios.

The original model introduced some parameters representing the vigilance state (i.e. active wakefulness vs slow-wave sleep). However, the present model only focused on the state of active wakefulness, since this is when memory-related theta-nested gamma oscillations occur. For all simulations, the parameters were set so that the network operated in the wakefulness regime in a healthy hippocampus (*Aussel et al., 2018*). The full expressions for all the parameters defined above can be found in *Table 2* for pyramidal cells and in *Table 3* for interneurons.

## Synaptic models

Inter-neuronal interactions were modeled as instantaneous AMPA and GABA-A synapses using the synaptic currents $I_{syn_E}$ and $I_{syn_I}$, respectively. Synaptic currents were described by the following bi-exponential differential equations:

$$
\begin{aligned}
I_{syn_{I,E}} &= g_{I,E}(V_m - E_{I,E}) \\
\frac{dg_{I,E}}{dt} &= \frac{1}{\tau_{g_{I,E}}}(-g_{I,E} + h_{I,E}) \\
\frac{dh_{I,E}}{dt} &= -h_{I,E}\frac{1}{\tau_{h_{I,E}}}
\end{aligned}
\tag{13}
$$

where $E_{I,E}$ are the synaptic resting potentials, and $\tau_{g_{I,E}}$ and $\tau_{h_{I,E}}$ are the synaptic time constants of rise and decay for inhibitory and excitatory neurons respectively. The occurrence of a pre-synaptic spike leads to an increase of the values $h_I$ or $h_E$ in the post-synaptic neuron by a fixed amount, which depends on the type of synapse and the region (due to the presence of cholinergic effects described in the initial model). Specific values for the intra-area and inter-area synaptic connections are given in *Table 4* and *Table 5*, respectively.

## Hippocampal formation: neuron types and numbers

Each area of the network is comprised of two populations, one excitatory and one inhibitory. Excitatory cells in the DG represent granule cells and pyramidal neurons in all other areas. Interneurons represent basket cells across all areas. The ratio between pyramidal neurons and interneurons was directly adapted from *Aussel et al., 2018*. The ratio between pyramidal neurons and interneurons was kept as a ratio of 10:1 for all areas except the dentate gyrus, where the ratio was 100:1. The number of neurons per subfield of the hippocampal formation is summarized in *Table 6*.

A two-dimensional simplified image depicting a coronal slice of the hippocampal formation (*Aussel et al., 2018*) was used as a basis for a two-dimensional manifold that was uniformly populated by neurons following a density-driven approach (*Rougier, 2018*). Pyramidal neurons were uniformly distributed within the stratum pyramidale (or within the stratum granulosum for the dentate gyrus) and interneurons were uniformly distributed within the stratum oriens. Initial neuron positions were drawn from a blue noise distribution and a Voronoi diagram was computed. To adjust the positions of the neurons over a centroidal Voronoi diagram, the Lloyd relaxation algorithm was applied for 1000 iterations. Transitioning from a two-dimensional manifold to a 3D reconstruction of the hippocampal formation was achieved through the addition of the third coordinate with values uniformly distributed between 0 and 15 mm.

## Hippocampal formation: inputs and connectivity

The hippocampal formation receives most of its external inputs from the EC. Activity from the EC is projected to all hippocampal subfields, starting with the DG. DG granule cells project onto the CA3 pyramidal cells via mossy cell fibers. CA3 projects in turn to CA1 via Schaffer collaterals. These connections form the tri-synaptic pathway. Direct connections from the EC towards the CA3 and CA1 subfields through the monosynaptic pathway are also considered. Pyramidal neurons from CA1 project to pyramidal neurons and interneurons in the EC, closing the hippocampal-entorhinal loop.

**Table 2.** Full list of parameter values and expressions for pyramidal neurons.

| Parameter | Expression |
|---|---|
| $A$ | $29.10^3 \, \mu\text{m}^2$ |
| $C_m$ | $1 \, \mu\text{F/cm}^2$ |
| $g_L$ | $0.01 \, \text{mS/cm}^2$ |
| $E_L$ | $-70 \, \text{mV}$ |
| $g_K$ | $5 \, \text{mS/cm}^2$ |
| $E_K$ | $-100 \, \text{mV}$ |
| $\alpha_{n,K}$ | $-0.032 \dfrac{V_m + 40 \, \text{mV}}{-1 + e^{-0.02 \, (V_m + 40 \, \text{mV})}}$ |
| $\beta_{n,K}$ | $0.5 \, e^{\frac{-(V_m + 45 \, \text{mV})}{40 \, \text{mV}}}$ |
| $g_{Na}$ | $50 \, \text{mS/cm}^2$ |
| $E_{Na}$ | $50 \, \text{mV}$ |
| $\alpha_{m,Na}$ | $-0.32 \dfrac{V_m + 42 \, \text{mV}}{e^{-\frac{V_m + 42 \, \text{mV}}{4 \, \text{mV}}} - 1}$ |
| $\beta_{m,Na}$ | $0.28 \dfrac{V_m + 15 \, \text{mV}}{e^{-\frac{V_m + 15 \, \text{mV}}{5 \, \text{mV}}} - 1}$ |
| $\alpha_{h,Na}$ | $0.128 \, e^{-\frac{V_m + 38 \, \text{mV}}{18 \, \text{mV}}}$ |
| $\beta_{h,Na}$ | $\dfrac{4}{1 + e^{-\frac{V_m + 15 \, \text{mV}}{5 \, \text{mV}}}}$ |
| $g_M$ | $90 \, \mu\text{S/cm}^2$ |
| $E_M$ | $-100 \, \text{mV}$ |
| $p_\infty$ | $\dfrac{1}{1 + e^{-0.01(V_m + 35 \, \text{mV})}}$ |
| $\tau_p$ | $\dfrac{1}{3.3 e^{\frac{(V_m + 35 \, \text{mV})}{20 \, \text{mV}}} + e^{-\frac{(V_m + 35 \, \text{mV})}{20 \, \text{mV}}}}$ |
| $g_{Ca}$ | $0.1 \, \text{mS/cm}^2$ |
| $E_{Ca}$ | $120 \, \text{mV}$ |
| $\alpha_{m,Ca}$ | $-0.055 \dfrac{V_m + 27 \, \text{mV}}{e^{-\frac{V_m + 27 \, \text{mV}}{17 \, \text{mV}}} - 1}$ |
| $\beta_{m,Ca}$ | $-0.94 e^{\frac{V_m + 75 \, \text{mV}}{17 \, \text{mV}}}$ |
| $\alpha_{h,Ca}$ | $-0.000457 e^{\frac{V_m + 13 \, \text{mV}}{50 \, \text{mV}}}$ |

*Table 2 continued on next page*

*Table 2 continued*

| Parameter | Expression |
|---|---|
| $\beta_{h,Ca}$ | $\dfrac{0.0065}{e^{-\frac{V_m+15\,\text{mV}}{28\,\text{mV}}}}$ |
| $g_{CAN}$ | $25\,\mu\text{S/cm}^2$ |
| $E_{CAN}$ | $-20\,\text{mV}$ |
| $\alpha_{m,CAN}$ | $0.0002\,e^{1.4\frac{[Ca]_i^{2+}}{0.5\,\text{mol/L}}}$ |
| $\beta_{m,CAN}$ | $0.0002\,e^{1.4}$ |

CA1 pyramidal neurons also project to the medial septum through the fornix. An overview of the connections between areas is presented in panel B of *Figure 1*.

In the current model, EC pyramidal neurons and interneurons receive oscillatory theta input from the medial septum in the form of an excitatory intracellular current as described in *Equation 2* and *Equation 5*. Projections from CA1 towards the medial septum were modeled as a signal representing the collective firing rate of the CA1 pyramidal neurons. All connections towards and from the medial septum are summarized in panels A and B of *Figure 1*.

Synaptic connectivity between neurons within a region is characterized using a probability $p$ and is distance-based, following a Gaussian-like distribution (*Equation 14*) with a width $\sigma$ of 2500 μm (excitatory synapses) and 350 μm (inhibitory synapses). The value of $A_{intra}$ defines the maximum

**Table 3.** Full list of parameter values and expressions for interneurons.

| Parameter | Expression |
|---|---|
| $A$ | $14.10^3\,\mu\text{m}^2$ |
| $C_m$ | $1\,\mu\text{F/cm}^2$ |
| $g_L$ | $0.1\,\text{mS/cm}^2$ |
| $E_L$ | $-65\,\text{mV}$ |
| $g_K$ | $9\,\text{mS/cm}^2$ |
| $E_K$ | $-90\,\text{mV}$ |
| $\alpha_{n,K}$ | $0.01\dfrac{V_m+34\,\text{mV}}{1-e^{-0.1(V_m+34\,\text{mV})}}$ |
| $\beta_{n,K}$ | $0.125e^{-\frac{V_m+44\,\text{mV}}{80\,\text{mV}}}$ |
| $g_{Na}$ | $35\,\text{mS/cm}^2$ |
| $E_{Na}$ | $55\,\text{mV}$ |
| $\alpha_{m,Na}$ | $0.1\dfrac{V_m+35\,\text{mV}}{1-e^{-0.1(V_m+35\,\text{mV})}}$ |
| $\beta_{m,Na}$ | $4e^{-\frac{V_m+60\,\text{mV}}{18\,\text{mV}}}$ |
| $\alpha_{h,Na}$ | $0.07e^{-\frac{V_m+58\,\text{mV}}{20\,\text{mV}}}$ |
| $\beta_{h,Na}$ | $\dfrac{1}{1+e^{-0.1(V_m+28\,\text{mV})}}$ |

**Table 4.** A pre-synaptic spike causes an increase in the conductances $h_e$ and $h_i$ in the post-synaptic neuron.

The values for the intra-area connections are given here. Empty cells indicate no connection between the populations.

|      | E→E   | E→I    | I→E     | I→I     |
|------|-------|--------|---------|---------|
| EC   |       | 20 pS  | 600 pS  |         |
| DG   |       | 180 pS | 1800 pS |         |
| CA3  | 20 pS | 20 pS  | 600 pS  |         |
| CA1  |       | 60 pS  | 1800 pS | 1800 pS |

probability of connection between two neurons separated by an infinitesimal distance (i.e. for $D \to 0$). The maximum values for $A_{intra}$ are given in **Table 7**.

$$p = A_{intra} \, \exp(-\frac{D^2}{2\sigma^2}) \tag{14}$$

Inter-area connectivity followed a similar Gaussian-like distribution (**Equation 15**), however, only excitatory projections were considered and the distance $D_z$ was computed only across the z-coordinate. Pyramidal neurons from the source area projected to pyramidal neurons and interneurons in the target area, with connectivity probabilities drawn from the said distribution with a width $\sigma$ of 1000 µm.

$$p = min(1, A_{inter} \exp(-\frac{D_z^2}{2\sigma^2})) \tag{15}$$

## Tuning input gain and connection strengths: targeted firing rates

We adjusted the input gain ($G_\theta$) and inter-area connection strengths ($A_{inter}$) to target an overall oscillatory rhythm $f_{osc,targ}$ at the driving frequency of the input, that is, 6 Hz, and a mean firing rate of each excitatory population $f_{exc,targ}$ also at 6 Hz (meaning that each excitatory cell should spike on average once per theta cycle). Because of the ratio between excitatory and inhibitory neurons in the model, this resulted in firing rates of about 60 Hz in inhibitory neurons. These targeted values were inspired by literature in behaving rodents showing that hippocampal pyramidal neurons typically fire at rates below 10 Hz, usually between 1 and 2 Hz, and that interneurons fire at rates between 20 and 80 Hz (**Hirase et al., 2001**). In practice, the obtained firing rates were constrained by the simplifications made in the model.

The mean population firing rate $f_{exc}$ was computed by counting the number of spikes in the last second of a 3 s simulation run, to avoid edge effects due to the non-physiological initial conditions, and by averaging this number over time and across neurons. The oscillatory frequency $f_{osc}$ was computed as the mean of the inverse of the timing between all pairs of consecutive peaks in the theta rhythm. Finally, the metric used to adjust parameters (input strength and connection strengths) was calculated as the Euclidean distance between the targeted and obtained firing rate and oscillatory rate:

**Table 5.** A pre-synaptic spike causes an increase in the conductances $h_e$ and $h_i$ in the post-synaptic neuron.

The values for the inter-area connections are given here. Empty cells indicate no connection between the areas. Recurrent projections are not allowed and are marked with dashes.

|        | Target |        |        |        |
|--------|--------|--------|--------|--------|
| Source | EC     | DG     | CA3    | CA1    |
| EC     | -      | 20 pS  | 20 pS  | 20 pS  |
| DG     |        | -      | 180 pS |        |
| CA3    |        |        | -      | 20 pS  |
| CA1    | 60 pS  |        |        | -      |

**Table 6.** Number of neurons per subfield of the hippocampal formation, divided by neuron type.

| Area | $N_{Exc}$ | $N_{Inh}$ |
|---|---|---|
| EC | 10,000 | 1,000 |
| DG | 10,000 | 100 |
| CA3 | 1,000 | 100 |
| CA1 | 10,000 | 1,000 |

$$J = \sqrt{(f_{exc} - f_{exc,targ})^2 + (f_{osc} - f_{osc,targ})^2} \tag{16}$$

## Tuning input and connection strengths: detailed procedure

The input gain ($G_\theta$, **Equation 2**) and inter-area connection strengths ($A_{inter}$, **Equation 15**) were sequentially adjusted using the following heuristics. All simulations were performed for a duration of 3 s, and the first 2 s were excluded from the analysis to avoid edge effects. The network was initialized with membrane voltages uniformly distributed in the range [-70, -60] mV. Our initial point was a fully uncoupled model in which all the connection strengths were set to 0. The tuning procedure was performed in the absence of noise.

1. First, we adjusted the amplitude of the theta input to a value that would produce a mean firing rate of 6 Hz in the excitatory population of the EC. To this end, the input strength was progressively increased from 0 to 0.5 nA in steps of 0.01 nA, and the value that maximized our metric was selected.
2. We then adjusted similarly the connection strength from the EC to the DG, increasing it from 0 to 20 in steps of 0.1.
3. We repeated the procedure to jointly tune the connection strengths from EC and DG to CA3, assuming an equal contribution from these two areas. These two connection strengths were set to the same value and were increased from 0 to 5 in steps of 0.1.
4. We followed the same procedure to jointly tune the connection strengths from CA3 and EC to CA1, assuming again an equal contribution from these two areas.
5. The last connection from CA1 to EC was more difficult to tune because it closes the feedback loop and induces complex dynamics when too strong. To tune it, we temporarily uncoupled CA1 from all other structures and temporarily added a fixed sinusoidal input at the target frequency of 6 Hz to both excitatory and inhibitory CA1 neurons. We first adjusted this input amplitude to generate a mean firing rate of about 6 Hz in CA1 excitatory neurons. Next, the connection strength from CA1 to EC was adjusted to also achieve a mean firing rate of 6 Hz in EC excitatory neurons.
6. Because EC receives both an external input and feedback from CA1, we assumed that both these contributions should be decreased for the system to behave in a physiological range in the presence of the feedback loop. We, therefore, decided to divide by two the connection strength from CA1 to EC obtained in step 5. All other connection strengths were then set again to the values tuned in the previous steps, and the temporary external input to CA1 was removed. The external theta input to the EC was reinstated and tuned again as in step 1.

**Table 7.** Maximum probability of connection ($A_{intra}$, **Equation 14**) between neurons within each region.

| | Py-Py | Py-Inh | Inh-Py | Inh-Inh |
|---|---|---|---|---|
| EC | 0 | 0.37 | 0.54 | 0 |
| DG | 0 | 0.06 | 0.14 | 0 |
| CA3 | 0.56 | 0.75 | 0.75 | 0 |
| CA1 | 0 | 0.28 | 0.3 | 0.7 |

**Table 8.** Inter-area connection strengths ($A_{inter}$).
The source is always the excitatory population of the subfield. The same values are used when targeting excitatory and inhibitory populations. Empty cells indicate no connections. EC: Entorhinal Cortex, DG: Dentate Gyrus.

| | Target | | | |
|---|---|---|---|---|
| Source | EC | DG | CA3 | CA1 |
| EC | - | 13.0 | 0.14 | 1.1 |
| DG | | - | 0.14 | |
| CA3 | | | - | 1.1 |
| CA1 | 0.2 | | | - |

Following the above procedure, the inter-area synaptic connectivity parameters were set for all subsequent simulations. The values are summarized in *Table 8*. Empty cells denote no effective connectivity.

## Numerical implementation

The model was implemented with the Brian2 libraries for Python (*Stimberg et al., 2019*; see Data availability section for access to the code). Simulations were performed using a timestep of 0.1ms and a total simulation duration ranging between 3 and 10 s (depending on the experiment, see results section). The average time for simulating 1 s for the complete model locally was 10 min.

## Neural mass model interfaced with Kuramoto oscillators

The neural masses represented in *Figure 3—figure supplement 4*, were modeled using the Wilson-Cowan formalism, with parameters adapted from *Onslow et al., 2014*. Specifically, the firing rates of the excitatory and inhibitory populations were determined by the following equations:

$$\begin{cases} \tau_E \dfrac{dE}{dt} & = -E + f(g_E\theta_E + W_{EE}E - W_{IE}I + stim_E(t)) \\ \tau_I \dfrac{dI}{dt} & = -I + f(g_I\theta_I + W_{EI}E) \end{cases} \tag{17}$$

with the following parameters: $\tau_E = \tau_I = 3.2ms$, $g_E = 0.7$, $g_I = 0$, $W_{EE} = 4.8$, $W_{EI} = W_{IE} = 4$, and $W_{II} = 0$. The sigmoid response function was not modified from the original work, and was defined as:

$$f(x) = \frac{1}{1 + e^{-\beta(x-x_m)}} \tag{18}$$

with parameters $\beta$ and $x_m$ set to 4 and 1 respectively, according to the original model. The Kuramoto oscillators were modeled according to *Equation 1*. For these simulations we used a set of $N = 100$ oscillators with a center frequency $f_0$ of 4 Hz, a synchronization ratio $\frac{k}{N}$ of 25, and a strong phase reset gain $G_{reset}$ of 90. Other parameters were kept as previously shown in *Table 1*. The neural masses were coupled to the Kuramoto oscillators by linking the variable $X(t)$ in *Equation 1* with the variable $E(t)$ in *Equation 17*.

We increased the phase reset gain significantly compared to the Hodgkin-Huxley model, as the Onslow model utilized a sigmoid function with values between 0 and 1, whereas the instantaneous firing rate of the populations of single-compartment neurons was much higher and therefore had a stronger phase resetting effect.

## Data analysis

During each simulation, we monitored and exported the following data for subsequent analysis: (i) spike timings per neuron, (ii) time series of ionic currents, (iii) septal input theta rhythm and phase, and (iv) time series of electrical stimulation. Theta phase was wrapped between $[-\pi, \pi]$ with a phase of 0 radians corresponding to the peak of theta rhythm. All PAC analyses were performed using the *TensorPAC* toolbox for Python (*Combrisson et al., 2020*).

## Firing rates

To obtain instantaneous firing rates, the corresponding spike trains were binned in 5 ms rectangular windows with a 90% overlap (i.e. consecutive windows were spaced by 0.5ms). The number of spikes within each bin was normalized by the bin size and the number of neurons in the group, yielding instantaneous population firing rates. Where reported (i.e. values $\mu_I$ and $\mu_E$ in **Figure 3A**), the mean population firing rates within a given time window (typically lasting several seconds) were computed by binning all spikes in that time window and normalizing by the window width and the number of neurons in the population.

## Spectral analyses

Power spectral density (PSD) estimates were calculated based on Welch's method. The periodogram was computed using 1 s windows with a 90% overlap, yielding a frequency resolution of 1 Hz. The average spectral power within a specific frequency band was calculated using Simpson's rule within the desired frequency band $\mu_I$. Spectrograms were computed using the short-time Fourier transform with a sliding Hann window of 100 ms width and 99% overlap, yielding a frequency resolution of 10 Hz.

## Modulation index

The MI (**Tort et al., 2008**) was used to estimate the degree of PAC between theta and gamma oscillations in the model. To compute the MI, the normalized firing rate traces were band-pass filtered in the frequency ranges of interest: 3–9 Hz for theta (referred to as the 'phase signal') and 40–80 Hz for gamma (referred to as the 'amplitude signal'). Then, the phase and amplitude time series were extracted from the filtered signals using the Hilbert transform. A histogram of the mean amplitude of gamma over the phase of theta was then extracted, using phase bins of 5 degrees. The MI was finally calculated as the Kullback-Leibler divergence between the mean amplitude distribution and the uniform distribution. A higher MI indicates stronger PAC. For a schematic representation regarding the computation of the MI, refer to Figure 1 in **Tort et al., 2010**. These computations were performed using the provided *PreferredPhase* and *Pac* methods from the *TensorPAC* library for Python (**Combrisson et al., 2020**).

## Comodulograms

Comodulograms represent the amount of PAC between two ranges of frequencies, used to extract respectively a phase and an amplitude signal. Specifically, we computed the MI between 80% overlapping 1 Hz frequency bands used to compute the phase of the signal (in the theta range), and 90% overlapping 10 Hz frequency bands used to compute the amplitude of the signal (in the gamma range). The resulting distribution of MI values was subsequently plotted as heat maps. An inherent problem with the computation of the MI using simulated data was the lack of frequency components in some frequency bands. Filtering the data within 1 Hz frequency bands thus created a flat signal, resulting in high values of the MI despite the absence of modulation. To overcome this limitation, we added uniform noise to the firing rate signals prior to computing the MI, with an amplitude of approximately 20% of the maximum instantaneous firing rates. The results with and without added noise are presented in **Figure 3—figure supplement 2**.

## Phase dependency of PAC

For a given theta frequency, PAC depends on the phase of the underlying theta oscillation. To identify this relationship and the theta phase that maximizes coupling, we used the *PreferredPhase* function of the *TensorPAC* toolbox. More precisely, we applied a Hilbert transform to extract the phase of the theta signal (firing rate band-pass filtered between 3 and 9 Hz) and the amplitude of the gamma signal band-pass filtered within narrow (10 Hz wide) frequency ranges between 20 and 100 Hz with a 90% overlap. For each narrow gamma range, we binned the amplitude with respect to the phase in a similar way as in the calculation of the MI. We obtained a vector of the binned high-frequency amplitudes with respect to the phase of the low-frequency phase, represented as a polar plot as in **Figure 3C**.

## Phase response curves

The PRC of an oscillatory system indicates the phase delay or advancement that follows a single pulse, as a function of the phase at which this input is delivered. To characterize the PRC of our computational model, we applied a single stimulation pulse to CA1 across different phases of the theta rhythm and calculated the resulting change in the theta phase. We split a single theta cycle into intervals of width $\pi/8$ radians and applied a single stimulation pulse of a given amplitude. For each case, we ran two simulations: one with a stimulation pulse and one without. Finally, we compared the theta phase 2.5ms post-stimulation and at the same time but in the absence of stimulation. The resulting values for the phase difference $\Delta\phi$ were plotted against the stimulation phase and are presented in *Figure 3D* for varying stimulation amplitudes.

## Acknowledgements

Simulations presented in this paper were carried out using the PlaFRIM experimental testbed, supported by Inria, CNRS (LABRI and IMB), Université de Bordeaux, Bordeaux INP and Conseil Régional d'Aquitaine (see https://www.plafrim.fr).

## Additional information

### Funding

| Funder | Grant reference number | Author |
| --- | --- | --- |
| Conseil Régional Aquitaine | Bordeaux Neurocampus chair of excellence | Fabien B Wagner |
| Université de Bordeaux | Bordeaux Neurocampus chair of excellence | Fabien B Wagner |
| European Research Council | ERC Starting Grant #101040391 (MEMOPROSTHETICS project) | Fabien B Wagner |

The funders had no role in study design, data collection and interpretation, or the decision to submit the work for publication.

### Author contributions

Nikolaos Vardalakis, Conceptualization, Software, Formal analysis, Investigation, Visualization, Methodology, Writing – original draft, Writing – review and editing; Amélie Aussel, Software, Formal analysis, Supervision, Investigation, Methodology, Writing – original draft, Writing – review and editing; Nicolas P Rougier, Conceptualization, Formal analysis, Supervision, Investigation, Methodology, Writing – original draft, Writing – review and editing; Fabien B Wagner, Conceptualization, Formal analysis, Supervision, Funding acquisition, Investigation, Methodology, Writing – original draft, Writing – review and editing

### Author ORCIDs

Nikolaos Vardalakis https://orcid.org/0000-0002-5436-4091
Amélie Aussel https://orcid.org/0000-0003-0498-2905
Nicolas P Rougier https://orcid.org/0000-0002-6972-589X
Fabien B Wagner http://orcid.org/0000-0002-9582-6109

Reviewer #1 (Public Review): https://doi.org/10.7554/eLife.87356.3.sa1
Reviewer #2 (Public Review): https://doi.org/10.7554/eLife.87356.3.sa2
Author response https://doi.org/10.7554/eLife.87356.3.sa3

## Additional files

### Supplementary files
• MDAR checklist

### Data availability

All model files and analysis scripts are available at the following repositories under the GNU General Public License v3.0 license on Zenodo (https://www.doi.org/10.5281/zenodo.8354987) and GitHub (https://github.com/NikVard/memstim-hh; copy archived at *Vardalakis, 2023*).

The following dataset was generated:

| Author(s) | Year | Dataset title | Dataset URL | Database and Identifier |
|---|---|---|---|---|
| Vardalakis N, Aussel A, Rougier NP, Wagner FB | 2023 | Dataset associated with Vardalakis et al. (2024), eLife | https://doi.org/10.5281/zenodo.8354987 | Zenodo, 10.5281/zenodo.8354987 |

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
