## [Editor Report · eLife assessment]

This study presents a computational model to explore how neurostimulation could impact hippocampal theta oscillations. The computational model combines a detailed physiologically realistic hippocampus model and an abstract theta oscillator. The study could provide **valuable** predictions on pathological changes in this network. The modelling is based on **convincing** approaches that could be improved with experimental validation in future experiments.

---

## [Referee Report · Reviewer #1 (Public Review)]

In this article, Vardakalis et al. propose a novel model of hippocampal oscillations whereby an external input (emulating the medial septum) can drive theta rhythms. This model displays phase-amplitude coupling of gamma oscillations, as well as theta resetting, which are known features of physiological theta that have been missing in previous models. The end goal proposed by the authors is to have a framework to explore the mechanisms of neurostimulation, which have shown promising applications in pathological conditions, but for which the underlying dynamics remain largely unknown. To reach this objective, the authors implement an existing biophysical model of the hippocampus that is able to generate gamma oscillations, and receives inputs from a set of Kuramoto oscillators to emulate theta drive originating from the medial septum.

Overall, the hypotheses and results are clearly presented and supported by high quality figures. The study is presented in a didactic way, making it easy for a broad audience to understand the significance of the results. The study does present some weaknesses that could easily be addressed by the authors. First, there are some anatomical inaccuracies: line 129 and fig1C, the authors omit medial septum projections to area CA1 (in addition to the entorhinal cortex). Moreover, in addition to CA1, CA3 also provides monosynaptic feedback projections to the medial septum CA3. Finally, an indirect projection from CA1/3 excitatory neurons to the lateral septum, which in turn sends inhibitory projections to the medial septum could be included or mentioned by the authors. This could be of particular relevance to support claims related to effects of neurostimulations, whereby minutious implementation of anatomical data could be key. If not updating their model, the authors could add this point to their limitation section, where they already do a good job of mentioning some limitations of using the EC as a sole oscillatory input to CA1. The authors test conditions of low theta inputs, which they liken to pathological states (line 112). It is not clear what pathology the authors are referring to, especially since a large amount of 'oscillopathies' in the septohippocampal system are associated with decreased gamma/PAC, but not theta oscillations (e.g. Alzheimer's disease conditions). While relevant for the clinical field, there is overall a missed opportunity to explain many experimental accounts with this novel model. Although to this day, clinical use of DBS is mostly restricted to electrical (and thus cell-type agnostic) stimulation, recent studies focusing on mechanisms of neurostimulations have manipulated specific subtypes in the medial septum and observed effects on hippocampal oscillations (e.g. see Muller & Remy, 2017 for review). Focusing stimulations in CA1 is of course relevant for clinical studies but testing mechanistic hypotheses by focusing stimulation on specific cell types could be highly informative. For instance, could the author reproduce recent optogenetic studies (e.g. Bender et al. 2015 for stimulation of fornix fibers; Etter et al., 2019 & Zutshi et al. 2018 for stimulation of septal inhibitory neurons)? Cell specific manipulations should at least be discussed by the authors.

Beyond these weaknesses, this study has a strong utility for researchers wanting to explore hypotheses in the field of neurostimulations. In particular, I see value in such models for exploring more intricate, phase specific effects of continuous, as well as close loop stimulations which are on the rise in systems neuroscience.

---

## [Referee Report · Reviewer #2 (Public Review)]

Theta-nested gamma oscillations (TNGO) play an important role in hippocampal memory and cognitive processes and are disrupted in pathology. Deep brain stimulation has been shown to affect memory encoding. To investigate the effect of pulsed CA1 neurostimulation on hippocampal TNGO the authors coupled a physiologically realistic model of the hippocampus comprising EC, DG, CA1, and CA3 subfields with an abstract theta oscillator model of the medial septum (MS). Pathology was modeled as weakened theta input from the MS to EC simulating MS neurodegeneration known to occur in Alzheimer's disease. The authors show that if the input from the MS to EC is strong (the healthy state) the model autonomously generates TNGO in all hippocampal subfields while a single neurostimulation pulse has the effect of resetting the TNGO phase. When the MS input strength is weaker the network is quiescent but the authors find that a single CA1 neurostimulation pulse can switch it into the persistent TNGO state, provided the neurostimulation pulse is applied at the peak of the EC theta. If the MS theta oscillator model is supplemented by an additional phase-reset mechanism a single CA1 neurostimulation pulse applied at the trough of EC theta also produces the same effect. If the MS input to EC is weaker still, only a short burst of TNGO is generated by a single neurostimulation pulse. The authors investigate the physiological origin of this burst and find it results from an interplay of CAN and M currents in the CA1 excitatory cells. In this case, the authors find that TNGO can only be rescued by a theta frequency train of CA1 pulses applied at the peak of the EC theta or again at either the peak or trough if the MS oscillator model is supplemented by the phase-reset mechanism.

The main strength of this model is its use of a fairly physiologically detailed model of the hippocampus. The cells are single-compartment models but do include multiple ion channels and are spatially arranged in accordance with the hippocampal structure. This allows the understanding of how ion channels (possibly modifiable by pharmacological agents) interact with system-level oscillations and neurostimulation. The model also includes all the main hippocampal subfields. The other strength is its attention to an important topic, which may be relevant for dementia treatment or prevention, which few modeling studies have addressed.

The work has several weaknesses. First, while investigations of hippocampal neurostimulation are important there are few experimental studies from which one could judge the validity of the model findings. All its findings are therefore predictions. It would be much more convincing to first show the model is able to reproduce some measured empirical neurostimulation effect before proceeding to make predictions. Second, the model is very specific. Or if its behavior is to be considered general it has not been explained why. For example, the model shows bistability between quiescence and TNGO, however what aspect of the model underlies this, be it some particular network structure or particular ion channel, for example, is not addressed. Similarly for the various phase reset behaviors that are found. We may wonder whether a different hippocampal model of TNGO, of which there are many published (for example [1-6]) would show the same effect under neurostimulation. This seems very unlikely and indeed the quiescent state itself shown by this model seems quite artificial. Some indication that particular ion channels, CAN and M are relevant is briefly provided and the work would be much improved by examining this aspect in more detail. In summary, the work would benefit from an intuitive analysis of the basic model ingredients underlying its neurostimulation response properties. Third, while the model is fairly realistic, considerable important factors are not included and in fact, there are much more detailed hippocampal models out there (for example [5,6]). In particular, it includes only excitatory cells and a single type of inhibitory cell. This is particularly important since there are many models and experimental studies where specific cell types, for example, OLM and VIP cells, are strongly implicated in TNGO. Other missing ingredients one may think might have a strong impact on model response to neurostimulation (in particular stimulation trains) include the well-known short-term plasticity between different hippocampal cell types and active dendritic properties. Fourth the MS model seems somewhat unsupported. It is modeled as a set of coupled oscillators that synchronize. However, there is also a phase reset mechanism included. This mechanism is important because it underlies several of the phase reset behaviors shown by the full model. However, it is not derived from experimental phase response curves of septal neurons of which there is no direct measurement. The work would benefit from the use of a more biologically validated MS model.

[1] Hyafil A, Giraud AL, Fontolan L, Gutkin B. Neural cross-frequency coupling: connecting architectures, mechanisms, and functions. Trends in neurosciences. 2015 Nov 1;38(11):725-40.

[2] Tort AB, Rotstein HG, Dugladze T, Gloveli T, Kopell NJ. On the formation of gamma-coherent cell assemblies by oriens lacunosum-moleculare interneurons in the hippocampus. Proceedings of the National Academy of Sciences. 2007 Aug 14;104(33):13490-5.

[3] Neymotin SA, Lazarewicz MT, Sherif M, Contreras D, Finkel LH, Lytton WW. Ketamine disrupts theta modulation of gamma in a computer model of hippocampus. Journal of Neuroscience. 2011 Aug 10;31(32):11733-43.

[4] Ponzi A, Dura-Bernal S, Migliore M. Theta-gamma phase-amplitude coupling in a hippocampal CA1 microcircuit. PLOS Computational Biology. 2023 Mar 23;19(3):e1010942.

[5] Bezaire MJ, Raikov I, Burk K, Vyas D, Soltesz I. Interneuronal mechanisms of hippocampal theta oscillations in a full-scale model of the rodent CA1 circuit. Elife. 2016 Dec 23;5:e18566.

[6] Chatzikalymniou AP, Gumus M, Skinner FK. Linking minimal and detailed models of CA1 microcircuits reveals how theta rhythms emerge and their frequencies controlled. Hippocampus. 2021 Sep;31(9):982-1002.

---

## [Author Response]

The following is the authors’ response to the original reviews.

Response to reviewers

We would like to thank the reviewers for their feedback. Below we address their comments and have indicated the associated changes in our point-by-point response (blue: answers, red: changes in manuscript).

**Reviewer #1:**
Overall, the hypotheses and results are clearly presented and supported by high quality figures. The study is presented in a didactic way, making it easy for a broad audience to understand the significance of the results. The study does present some weaknesses that could easily be addressed by the authors.

We thank the reviewer for appreciating our work and providing useful suggestions for improvement.

1. First, there are some anatomical inaccuracies: line 129 and fig1C, the authors omit m.dial septum projections to area CA1 (in addition to the entorhinal cortex). Moreover, in addition to CA1, CA3 also provides monosynaptic feedback projections to the medial septum CA3. Finally, an indirect projection from CA1/3 excitatory neurons to the lateral septum, which in turn sends inhibitory projections to the medial septum could be included or mentioned by the authors. This could be of particular relevance to support claims related to effects of neurostimulations, whereby minutious implementation of anatomical data could be key.If not updating their model, the authors could add this point to their limitation section, where they already do a good job of mentioning some limitations of using the EC as a sole oscillatory input to CA1.

We acknowledge that our current model strongly simplifies the interconnections between the medial septum and the hippocampal formation, but including more anatomical details is beyond the scope of this manuscript and would be a topic for future work. Nevertheless, we followed the reviewer’s advice to stress this point in our manuscript. First, we moved a paragraph that was initially in the “methods” section to the “results” section (L.141-150 of the revised manuscript):

“Biologically, GABAergic neurons from the medial septum project to the EC, CA3, and CA1 fields of the hippocampus (Toth et al., 1993; Hajós et al., 2004; Manseau et al., 2008; Hangya et al., 2009; Unal et al., 2015; Müller and Remy, 2018). Although the respective roles of these different projections are not fully understood, previous computational studies have suggested that the direct projection from the medial septum to CA1 is not essential for the production of theta in CA1 microcircuits (Mysin et al., 2019). Since our modeling of the medial septum is only used to generate a dynamic theta rhythm, we opted for a simplified representation where the medial septum projects only to the EC, which in turn drives the different fields of the hippocampus. In our model, Kuramoto oscillators are therefore connected to the EC neurons and they receive projections from CA1 neurons (see methods for more details).”

Second, we expanded the corresponding paragraph in the limitation section to discuss this point further (L.398-415 of the revised manuscript):

“We decided to model septal pacemaker neurons projecting to the EC as the main source of hippocampal theta as reported in multiple experimental studies (Buzsáki, 2002; Buzsáki et al., 2003; Hangya et al., 2009). However, experimental findings and previous models have also proposed that direct septal inputs are not essential for theta generation (Wang, 2002; Colgin et al., 2013; Mysin et al., 2019), but play an important role in phase synchronization of hippocampal neurons. Furthermore, the model does not account for the connections between the lateral and medial septum and the hippocampus (Takeuchi et al., 2021). These connections include the inhibitory projections from the lateral to the medial septum and the monosynaptic projections from the hippocampal CA3 field to the lateral septum. An experimental study has highlighted the importance of the lateral septum in regulating the hippocampal theta rhythm (Bender et al., 2015), an area that has not been included in the model. Specifically, theta-rhythmic optogenetic stimulation of the axonal projections from the lateral septum to the hippocampus was shown to entrain theta oscillations and lead to behavioral changes during exploration in transgenic mice. To account for these discrepancies, our model could be extended by considering more realistic connectivity patterns between the medial / lateral septum and the hippocampal formation, including glutamatergic, cholinergic, and GABAergic reciprocal connections (Müller and Remy, 2018), or by considering multiple sets of oscillators each representing one theta generator.”

2. The authors test conditions of low theta inputs, which they liken to pathological states (line 112). It is not clear what pathology the authors are referring to, especially since a large amount of 'oscillopathies' in the septohippocampal system are associated with decreased gamma/PAC, but not theta oscillations (e.g. Alzheimer's disease conditions).

In the manuscript, we referred to “oscillopathies” in a broad sense way as we did not want to overstate the biological implications of the model or the way we modeled pathological states. To our knowledge, several studies have yielded inconsistent results regarding the specific changes in theta or gamma power in Alzheimer’s disease, and the most convincing alteration seems to be the theta-gamma phase-amplitude coupling (PAC) (for review see e.g., Kitchigina, V. F. Alterations of Coherent Theta and Gamma Network Oscillations as an Early Biomarker of Temporal Lobe Epilepsy and Alzheimer’s Disease. Front Integr Neurosci 12, 36 (2018)), as also mentioned by the reviewer.

In this study, the most straightforward way to reduce theta-gamma PAC was to reduce the amplitude of the oscillators’ gain, which affected theta power, gamma power, and theta-gamma PAC (Figure 5 of the revised manuscript). Affecting their synchronization level (i.e., the order parameter) did not affect any of these variables (Figure 5 – Figure Supplement 4).

In order to alter theta-gamma PAC without affecting theta or gamma power, we believe that more complex changes should be performed in the model, likely at the level of individual neurons in the hippocampal formation. For example, cholinergic deprivation has been previously used in a multi-compartment model of the hippocampal CA3 to mimic Alzheimer’s disease and to draw functional implications on the slowing of theta oscillations and the storage of new information (Menschik, E. D. & Finkel, L. H. Neuromodulatory control of hippocampal function: towards a model of Alzheimer’s disease. Artif Intell Med 13, 99–121 (1998)).

This has now been added to the limitations section (L.458-465 of the revised manuscript):

“Finally, we likened conditions of low theta input to pathological states characteristic of oscillopathies such as Alzheimer’s disease, as these conditions disrupted all aspects of theta-gamma oscillations in our model: theta power, gamma power, and theta-gamma PAC (Figure 5). However, it should be noted that changes in theta or gamma power in these pathologies are often unclear, and that the most consistent alteration that has been reported in Alzheimer’s disease is a reduction of theta-gamma PAC (for review, see Kitchigina, 2018). Future work should explore the effects of cellular alterations intrinsic to the hippocampal formation and their impact on theta-gamma oscillations.”

3. While relevant for the clinical field, there is overall a missed opportunity to explain many experimental accounts with this novel model. Although to this day, clinical use of DBS is mostly restricted to electrical (and thus cell-type agnostic) stimulation, recent studies focusing on mechanisms of neurostimulations have manipulated specific subtypes in the medial septum and observed effects on hippocampal oscillations (e.g. see Muller & Remy, 2017 for review). Focusing stimulations in CA1 is of course relevant for clinical studies but testing mechanistic hypotheses by focusing stimulation on specific cell types could be highly informative. For instance, could the author reproduce recent optogenetic studies (e.g. Bender et al. 2015 for stimulation of fornix fibers; Etter et al., 2019 & Zutshi et al. 2018 for stimulation of septal inhibitory neurons)? Cell specific manipulations should at least be discussed by the authors.

We acknowledge the importance of cell-type-specific manipulation in the septo-hippocampal circuitry. However, our model was designed to study neurostimulation protocols that affect the hippocampal formation, not the medial septum, which is why only the hippocampal formation is composed of biophysically realistic (i.e., conductance-based) neuronal models. To replicate the various studies mentioned by the reviewer (which are all very relevant), we would need to implement a biophysical model of the medial septum, which would be an entirely new project.

Nevertheless, we can use the existing model to replicate optogenetic studies that induced gamma oscillations in excitatory-inhibitory circuits, using either ramped photostimulation targeting excitatory neurons (Adesnik et al., 2010; Akam et al., 2012; Lu et al., 2015), or pulsed stimulation driving inhibitory cells in the gamma range (Cardin et al., 2009; Iaccarino et al., 2016). In fact, such approaches have been demonstrated not just in the hippocampus but also in the neocortex, and represent a hallmark of local excitatory-inhibitory circuits. To account for these experimental results and replicate them, we have added 4 new figures (Figure 2 and its 3 figure supplements) and an extensive section in the results part (L.151-217 of the revised manuscript):

“From a conceptual point of view, our model is thus composed of excitatory-inhibitory (E-I) circuits connected in series, with a feedback loop going through a population of coupled phase oscillators. In the next sections, we first describe the generation of gamma oscillations by individual E-I circuits (Figure 2), and illustrate their behavior when driven by an oscillatory input such as theta oscillations (Figure 3). We then present a thorough characterization of the effects of theta input and stimulation amplitude on theta-nested gamma oscillations (Figure 4 and Figure 5). Finally, we present some results on the effects of neurostimulation protocols for restoring theta-nested gamma oscillations in pathological states (Figure 6 and Figure 7).

Generation of gamma oscillations by E-I circuits

It is well-established that a network of interconnected pyramidal neurons and interneurons can give rise to oscillations in the gamma range, a mechanism termed pyramidal-interneuronal network gamma (PING) (Traub et al., 2004; Onslow et al., 2014; Segneri et al., 2020;). This mechanism has been observed in several optogenetic studies with gradually increasing light intensity (i.e., under a ramp input) affecting multiple different circuits, such as layer 2-3 pyramidal neurons of the mouse somatosensory cortex (Adesnik et al., 2010), the CA3 field of the hippocampus in rat in vitro slices (Akam et al., 2012), and in the non-human primate motor cortex (Lu et al., 2015). In all cases, gamma oscillations emerged above a certain threshold in terms of photostimulation intensity, and the frequency of these oscillations was either stable or slightly increased when increasing the intensity further. We sought to replicate these findings with our elementary E-I circuits composed of single-compartment conductance-based neurons driven by a ramping input current (Figure 2 and Figure S2). As an example, all the results in this section will be shown for an E-I circuit that has similar connectivity parameters as the CA1 field of the hippocampus in our complete model (see section “Hippocampal formation: inputs and connectivity” in the methods).

For low input currents provided to both neuronal populations, only the highly-excitable interneurons were activated (Figure 2A). For a sufficiently high input current (i.e., a strong input that could overcome the inhibition from the fast-spiking interneurons), the pyramidal neurons started spiking as well. As the amplitude of the input increased, the activity of the both neuronal populations became synchronized in the gamma range, asymptotically reaching a frequency of about 60 Hz (Figure 2A bottom panel). Decoupling the populations led to the abolition of gamma oscillations (Figure 2B), as neuronal activity was determined solely by the intrinsic properties of each cell. Interestingly, when the ramp input was provided solely to the excitatory population, we observed that the activity of the pyramidal neurons preceded the activity of the inhibitory neurons, while still preserving the emergence of gamma oscillations (Figure S2 A). As expected, decoupling the populations also abolished gamma oscillations, with the excitatory neurons spiking a frequency determined by their intrinsic properties and the inhibitory population remaining silent (Figure S2B).

To further characterize the intrinsic properties of individual inhibitory and excitatory neurons, we derived their input-frequency (I-F) curves, which represent the firing rate of individual neurons in response to a tonic input (Figure S3A). We observed that for certain input amplitudes, the firing rates of both types of neurons was within the gamma range. Interestingly, in the absence of noise, each population could generate by itself gamma oscillations that were purely driven by the input and determined by the intrinsic properties of the neurons (Figure S3B). Adding stochastic Gaussian noise in the membrane potential disrupted these artificial oscillations in decoupled populations (Figure S3C). All subsequent simulations were run with similar noise levels to prevent the emergence of artificial gamma oscillations.

Another potent way to induce gamma oscillations is to drive fast-spiking inhibitory neurons using pulsed optogenetic stimulation at gamma frequencies, a strategy that has been used both in the neocortex (Cardin et al., 2009) and hippocampal CA1 (Iaccarino et al., 2016). In particular, Cardin and colleagues systematically investigated the effect of driving either excitatory or fast-spiking inhibitory neocortical neurons at frequencies between 10 and 200 Hz (Cardin et al., 2009). They showed that fast-spiking interneurons are preferentially entrained around 40-50 Hz, while excitatory neurons respond better to lower frequencies. To verify the behavior of our model against these experimental data, we simulated pulsed optogenetic stimulation as an intracellular current provided to our reduced model of a single E-I circuit. Stimulation was applied at frequencies between 10 and 200 Hz to excitatory cells only, to inhibitory cells only, or to both at the same time (Figure S4). The population firing rates were used as a proxy for the local field potentials (LFP), and we computed the relative power in a 10-Hz band centered around the stimulation frequency, similarly to the method proposed in (Cardin et al., 2009). When presented with continuous stimulation across a range of frequencies in the gamma range, interneurons showed the greatest degree of gamma power modulation (Figure S4). Furthermore, when the stimulation was delivered to the excitatory population, the relative power around the stimulation frequency dropped significantly in frequencies above 10 Hz, similar to the reported experimental data (Cardin et al., 2009). The main difference between our simulation results and these experimental data is the specific frequencies at which fast-spiking interneurons showed resonance, which was slow gamma around 40 Hz in the mouse barrel cortex and fast gamma around 90 Hz in our model. This could be attributed to several factors, such as differences in the cellular properties between cortical and hippocampal fast-spiking interneurons, or the differences between the size of the populations and their relevant connectivity in the cortex and the hippocampus.”

**Author response image 1. sa3fig1:** Figure 2. Emergence of gamma oscillations in coupled excitatory-inhibitory populations under ramping input to both populations. (**A**) Two coupled populations of excitatory pyramidal neurons (NE = 1000) and inhibitory interneurons (NI = 100) are driven by a ramping current input (0 nA to 1 nA) for 5 s. As the input becomes stronger, oscillations start to emerge (shaded green area), driven by the interactions between excitatory and inhibitory populations. The green inset shows the raster plot (neuronal spikes across time) of the two populations during the green shaded period (red for inhibitory; blue for excitatory). When the input becomes sufficiently strong (shaded magenta area), the populations become highly synchronized and produce oscillations in the gamma range (at approximately 50 Hz). The spectrogram (bottom panel) shows the power of the instantaneous firing rate of the pyramidal population as a function of time and frequency. It reveals the presence of gamma oscillations that emerge around 2s and increase in frequency until 4 s, when they settle at approximately 60 Hz. (**B**) Similar depiction as in panel A. with the pyramidal-interneuronal populations decoupled. The absence of coupling leads to the abolition of gamma oscillations, each cell spiking activity being driven by its own inputs and intrinsic properties.

**Author response image 2. sa3fig2:** Figure S2 (Figure 2 – Figure Supplement 1). Emergence of gamma oscillations in coupled excitatoryinhibitory populations under ramping input to the excitatory population. Similar representation as in Figure 2, but with the input provided only to the excitatory population. All conclusions remain the same. In addition, the inhibitory population does not show any spiking activity in the decoupled case.

**Author response image 3. sa3fig3:** Figure S3 (Figure 2 – Figure Supplement 2). Cell-intrinsic spiking activity in decoupled excitatory and inhibitory populations under ramping input. (**A**) Input-Frequency (I-F) curves for excitatory cells (left panel; pyramidal neurons with ICAN) and inhibitory cells (right panel; interneurons, fast-spiking) used in the model. Above a certain tonic input (around 0.35 nA for excitatory and 0.1 nA for inhibitory neurons), neurons can spike in the gamma range. (**B**) Raster plot showing the spiking activity of excitatory (blue, NE = 1000) and inhibitory (red, NI = 100) neurons in decoupled populations under ramping input (top trace) and in the absence of noise in the membrane potential. Despite random initial conditions across neurons, oscillations emerge in both populations due to the intrinsic properties of the cells, with a frequency that is predicted by the respective I-F curves (panel A.). (**C**) Similar representation as panel B. but with the addition of stochastic noise in the membrane potential of each neuron. The presence of noise disrupts the emergence of oscillations in these decoupled populations.

**Author response image 4. sa3fig4:** Figure S3 (Figure 2 – Figure Supplement 2). Cell-intrinsic spiking activity in decoupled excitatory and inhibitory populations under ramping input. (**A**) Input-Frequency (I-F) curves for excitatory cells (left panel; pyramidal neurons with ICAN) and inhibitory cells (right panel; interneurons, fast-spiking) used in the model. Above a certain tonic input (around 0.35 nA for excitatory and 0.1 nA for inhibitory neurons), neurons can spike in the gamma range. (**B**) Raster plot showing the spiking activity of excitatory (blue, NE = 1000) and inhibitory (red, NI = 100) neurons in decoupled populations under ramping input (top trace) and in the absence of noise in the membrane potential. Despite random initial conditions across neurons, oscillations emerge in both populations due to the intrinsic properties of the cells, with a frequency that is predicted by the respective I-F curves (panel A.). (**C**) Similar representation as panel B. but with the addition of stochastic noise in the membrane potential of each neuron. The presence of noise disrupts the emergence of oscillations in these decoupled populations.

Beyond these weaknesses, this study has a strong utility for researchers wanting to explore hypotheses in the field of neurostimulations. In particular, I see value in such models for exploring more intricate, phase specific effects of continuous, as well as close loop stimulations which are on the rise in systems neuroscience.

We thank the reviewer for this appreciation of our work and its future perspectives.

**Recommendations For The Authors:**
Line 144, the authors mention that their MI values are erroneous in absence of additive noise - could this be due to the non-sinusoidal nature of the phase signal recorded, and be fixed by upscaling model size?

We thank the reviewer for this question and suggestion. The main reason behind the errors in the computation of the MI lies in the complete absence of oscillations at specific frequencies. Filtered signals within specific bands produced a power of 0 (or extremely low values), as seen in the power spectral densities. In such cases, the phase signal was not mathematically defined, but the toolbox we used to compute it still returned a numerical result that was inaccurate (for more details on the computation of the MI see Tort et al., 2010). To mitigate this numerical artefact, we decided to add uniform noise in the computed firing rates. This strategy is illustrated on Figure S6 (Figure 3 – Figure Supplement 2), which we have copied below for reference. Alternative approaches could probably have been used, such as increasing the noise in the membrane potential so that neurons would start spiking with firing rates that show more realistic power spectra, even in the absence of external inputs.

**Author response image 5. sa3fig5:** Figure S6 (Figure 3 – Figure Supplement 2). Quantification of PAC with and without noise. (**A**) Quantifying PAC in the absence of noise produced inaccurate identification of the coupled frequency bands, due to the complete absence of oscillations at some frequencies. All analyses are based on the CA1 firing rates (top traces) during a representative simulation. Power spectral densities of these firing rates (left) indicate that some frequencies have a power of 0. PAC of the excitatory population was assessed using two graphical representations, the polar plot (middle) and comodulogram (right), and quantified using the MI. The comodulogram was calculated by computing the MI across 80% overlapping 1-Hz frequency bands in the theta range and across 90% overlapping 10-Hz frequency bands in the gamma range and subsequently plotted as a heat map. In the absence of noise, a slow theta frequency centered around 5 Hz is found to modulate a broad range of gamma frequencies between 40 and 100 Hz. The value indicated on the comodulogram indicates the average MI in the 3-9 Hz theta range and 40-80 Hz gamma range. As in Figure 2, the polar plot represents the amplitude of gamma oscillations (averaged across all theta cycles) at each phase of theta (theta range: 3-9 Hz, phase indicated as angular coordinate) and for different gamma frequencies (radial coordinate, binned in 1-Hz ranges). (**B**) Adding uniform noise to the firing rate (with an amplitude ranging between 15 and 25% of the maximum firing rate) improved the identification of the coupled frequency bands. In this case, the slower theta frequency centered around 5 Hz modulates a gamma band located between 45 and 75 Hz.

**Reviewer #2:**
The main strength of this model is its use of a fairly physiologically detailed model of the hippocampus. The cells are single-compartment models but do include multiple ion channels and are spatially arranged in accordance with the hippocampal structure. This allows the understanding of how ion channels (possibly modifiable by pharmacological agents) interact with system-level oscillations and neurostimulation. The model also includes all the main hippocampal subfields. The other strength is its attention to an important topic, which may be relevant for dementia treatment or prevention, which few modeling studies have addressed. The work has several weaknesses.

We thank the reviewer for appreciating our detailed description of the hippocampal formation and the focus on neurostimulation applications that aim at treating oscillopathies, especially dementia.

1. First, while investigations of hippocampal neurostimulation are important there are few experimental studies from which one could judge the validity of the model findings. All its findings are therefore predictions. It would be much more convincing to first show the model is able to reproduce some measured empirical neurostimulation effect before proceeding to make predictions.

We acknowledge that the results presented in Figures 4-7 of the revised manuscript cannot be compared to existing experimental data, and are therefore purely predictive. Future experimental work is needed to verify these predictions.

Yet, we would also like to stress that the motivation behind this project was the inadequacy of previous models of theta-nested gamma oscillations (Onslow et al., 2014; Aussel et al., 2018; Segneri et al., 2020) to account for the mechanism of theta phase reset that occurs during electrical stimulation of the fornix or perforant path (Williams and Givens, 2003). Since we could not use these previous models to study the effects of neurostimulation on theta-nested gamma oscillations, we had to modify them to account for a dynamical theta input, which is the main methodological novelty that is reported in our manuscript (Figures 1 and 3 of the revised manuscript).

Despite the scarcity of experimental studies that could confirm the full model, we sought to replicate a few experimental findings that employed optogenetic stimulation to induce gamma oscillations in individual excitatory-inhibitory circuits. Although not specific to the hippocampus, these studies have shown that gamma oscillations can be induced using either ramped photostimulation targeting excitatory neurons (Adesnik et al., 2010; Akam et al., 2012; Lu et al., 2015), or pulsed stimulation driving inhibitory cells in the gamma range (Cardin et al., 2009; Iaccarino et al., 2016). To account for these experimental results and replicate them, we have added 4 new figures (Figure 2 and its 3 figure supplements) and an extensive section in the results part (L.141-217 of the revised manuscript). The added section and related figures are indicated in our response to reviewer 1, comment 3 (p 2-7).

2.1. Second, the model is very specific. Or if its behavior is to be considered general it has not been explained why.

Although the spatial organization and cellular details of the model are indeed very specific, its general behavior, i.e., the production of theta-nested gamma oscillations and theta phase reset, are common to any excitatory-inhibitory circuit interconnected with Kuramoto oscillators. To illustrate this point, we have generalized our approach to the neural mass model developed by Onslow and colleagues (Onslow ACE, Jones MW, Bogacz R. A Canonical Circuit for Generating Phase-Amplitude Coupling. PLoS ONE. 2014 Aug; 9(8):e102591). These results are represented in a new supplementary figure (Figure3 – Figure Supplement 4), and briefly described in a new paragraph of the results section (L.262-268 of the revised manuscript):

“Importantly, our approach is generalizable and can be applied to other models producing theta-nested gamma oscillations. For instance, we adapted the neural mass model by Onslow and colleagues (Onslow et al., 2014), replaced the fixed theta input by a set of Kuramoto oscillators, and demonstrated that it could also generate theta phase reset in response to single-pulse stimulation (Figure S8). These results illustrate that the general behavior of our model is not specific to the tuning of individual parameters in the conductancebased neurons, but follows general rules that are captured by the level of abstraction of the Kuramoto formalism.”

**Author response image 6. sa3fig6:** Figure S8 (Figure 3 – Figure Supplement 4). A neural mass model of coupled excitatory and inhibitory neurons driven by Kuramoto oscillators generates theta-nested gamma oscillations and theta phase reset. (**A**) Two coupled neural masses (one excitatory and one inhibitory) driven by Kuramoto oscillators, which represent a dynamical oscillatory drive in the theta range, were used to implement a neural mass equivalent to our conductance-based model represented in Figure 1. Neural masses were modeled using the WilsonCowan formalism, with parameters adapted from Onslow et al. (2014) (WEE = 4.8, WEI = WIE = 4, WII = 0). (**B**) The normalized population firing rates exhibit theta-nested gamma oscillations (middle and bottom panels) in response to the dynamic theta rhythm (top panel). A stimulation pulse delivered at the descending phase of the rhythm to both populations (marked by the inverted red triangle) produces a robust theta phase reset, similarly to Figure 3A.

This simplified model is described in more details in the methods (L.694-710 of the revised manuscript). Additionally, the generation of gamma oscillations by individual excitatory-inhibitory circuits is now described in details in the added section “Generation of gamma oscillations by E-I circuits” (L.159-217 of the revised manuscript), which has already been discussed in our response to reviewer 1, comment 3 (p 2-7).

2.2. For example, the model shows bistability between quiescence and TNGO, however what aspect of the model underlies this, be it some particular network structure or particular ion channel, for example, is not addressed.

We thank the reviewer for mentioning this point, which we have now addressed. The “bistable” behavior that we reported occurs for values of the theta input that are just below the threshold to induce selfsustained theta-gamma oscillations (Figure 5 of the revised manuscript, point B). Moreover, the presence of the Calcium-Activated-Nonspecific (CAN) cationic channel, which is expressed by pyramidal neurons in the entorhinal cortex, CA3, and CA1 fields of the hippocampus, is necessary for this behavior to occur. Indeed, abolishing CAN channels in all areas of the model suppresses this behavior. We have now addressed this point in a new supplementary figure (Figure 5 – Figure Supplement 4) and a short description in the text (L.287-303 of the revised manuscript).

“In the presence of dynamic theta input, the effects of single-pulse stimulation depended both on theta input amplitude and stimulation amplitude, highlighting different regimes of network activity (Figure 5 and Figure S9, Figure S10, Figure S11). For low theta input, theta-nested gamma oscillations were initially absent and could not be induced by stimulation (Figure 5A). At most, the stimulation could only elicit a few bursts of spiking activity that faded away after approximately 250 ms, similar to the rebound of activity seen in the absence of theta drive. For increasing theta input, the network switched to an intermediate regime: upon initialization at a state with no spiking activity, it could be kicked to a state with self-sustained theta-nested gamma oscillations by a single stimulation pulse of sufficiently high amplitude (Figure 5B). This regime existed for a range of septal theta inputs located just below the threshold to induce self-sustained theta-gamma oscillations without additional stimulation, as characterized by the post-stimulation theta power, gamma power, and theta-gamma PAC (Figure 5D). Removing CAN currents from all areas of the model abolished this behavior (Figure S12), which is interesting given the role of this current in the multistability of EC neurons (Egorov et al., 2002; Fransen et al., 2006) and in the intrinsic ability of the hippocampus to generate thetanested gamma oscillations (Giovannini et al., 2017). For the highest theta input, the network became able to spontaneously generate theta-nested gamma oscillations, even when initialized at a state with no spiking activity and without additional neurostimulation (Figure 5C).”

**Author response image 7. sa3fig7:** Figure S12 (Figure 5 – Figure Supplement 4). CAN currents are necessary for the production of selfsustained theta-gamma oscillations in response to single-pulse stimulation. (**A**) Same as Figure 5B. (**B**) Similar simulation as panel A., but without the presence of CAN currents in the EC, CA3 and CA1 fields of the hippocampus. Removing CAN currents from the model abolishes self-sustained theta-nested gamma oscillations in response to a single stimulation pulse (for the parameters represented in Figure 5, point B).

Furthermore, we realized that the terminology “bistable” may not be justified as we could not perform a systematic bifurcation analysis, which is typically carried out in simpler neural mass models (e.g., Onslow et al., 2014; Segneri et al., 2020). Therefore, we decided to rephrase the sentences about “bistability” to keep a more general terminology. The following sentences were revised:

L.20-23: “We showed that, for theta inputs just below the threshold to induce self-sustained theta-nested gamma oscillations, a single stimulation pulse could switch the network behavior from non-oscillatory to a state producing sustained oscillations.”

L.305-309: “Based on the above analyses, we considered two pathological states: one with a moderate theta input (i.e., moderately weak projections from the medial septum to the EC) that allowed the initiation of selfsustained oscillations by single stimulation pulses (Figure 5, point B), and one with a weaker theta input characterized by the complete absence of self-sustained oscillations even following transient stimulation(Figure 5, point A).”

L.316-317: “In the case of a moderate theta input and in the presence of phase reset, delivering a pulse at either the peak or trough of theta could induce theta-nested gamma oscillations (Figure 6A and 6C).”

L.353-357: “A very interesting finding concerns the behavior of the model in response to single-pulse stimulation for certain values of the theta amplitude (Figure5). For low theta amplitudes, a single stimulation pulse was capable of switching the network behavior from a state with no spiking activity to one with prominent theta-nested gamma oscillations. Whether such an effect can be induced in vivo in the context of memory processes remains an open question.”

2.3. Similarly for the various phase reset behaviors that are found.

We would like to clarify the fact that the observed phase reset curves (reported in Figure 3D) are a direct consequence of the choice of an appropriate phase response function for the Kuramoto oscillators representing the medial septum. This choice is inspired by experimentally measured phase response curves from CA3 neurons. These aspects are described briefly in the introduction and in more details in the methods, as indicated below:

L.101: “This new hybrid dynamical model could generate both theta-nested gamma oscillations and theta phase reset, following a particular phase response curve (PRC) inspired by experimental literature (Lengyel et al., 2005; Akam et al., 2012; Torben-Nielsen et al., 2010).”

L.528-537: “Hereafter, we call the term Z(θ) the phase response function, to distinguish it from the PRC obtained from experimental data or simulations (see section below "Data Analysis", "Phase Response Curve"). Briefly, the PRC of an oscillatory system indicates the phase delay or advancement that follows a single pulse, as a function of the phase at which this input is delivered. The phase response function Z(θ) was chosen to mimic as well as possible experimental PRCs reported in the literature (Lengyel et al., 2005; Kwag and Paulsen, 2009; Akam et al., 2012). These PRCs appear biphasic and show a phase advancement (respectively delay) for stimuli delivered in the ascending (respectively descending) slope of theta. To accurately model this behavior, we used the following equation for the phase response function, where ϕpeak represents the phase at which the theta rhythm reaches its maximum and the parameter ϕoffset controls the desired phase offset from the peak:

**Author response image 8. sa3fig8:** On the figure below, we illustrate the phase response curves of CA3 neurons measured by Lengyel et al., 2005 (**A**), and compare it with our simulated phase response curves (**B**). Note that the conventions for phase advance and phase delay are reversed between the two panels.

Finally, we would like to acknowledge that the model “is not derived from experimental phase response curves of septal neurons of which there is no direct measurement”, as mentioned by the reviewer in their comment 4 below. Despite the lack of experimental data specific to medial septum neurons, we argue that this phase response function is the only one that mathematically supports the generation of self-sustained theta-nested gamma oscillations in our current model. This statement is illustrated by Figure S7 (Figure 3 – Figure Supplement 3) and is mentioned in the results (L.249-261 of the revised manuscript):

We modeled this behavior by a specific term (which we called the phase response function) in the general equation of the Kuramoto oscillators (see methods, Equation 1). Importantly, introducing a phase offset in the phase response function disrupted theta-nested gamma oscillations (Figure S7), which suggests that the septohippocampal circuitry must be critically tuned to be able to generate such oscillations. The strength of phase reset could also be adjusted by a gain that was manually tuned. In the presence of the physiological phase response function and of a sufficiently high reset gain, a single stimulation pulse delivered to all excitatory and inhibitory CA1 neurons could reset the phase of theta to a value close to its peaks (Figure 3A). We computed the PRC of our simulated data for different stimulation amplitudes and validated that our neuronal network behaved according to the phase response function set in our Kuramoto oscillators (Figure 3D). It should be noted that including this phase reset mechanism affected the generated theta rhythm even in the absence of stimulation, extending the duration of the theta peak and thereby slowing down the frequency of the generated theta rhythm.

**Author response image 9. sa3fig9:** Figure S7 (Figure 3 – Figure Supplement 3). Network behavior generated by Kuramoto oscillators with nonphysiological phase response functions. Each panel is similar to Figure 3A, but with a different offset added to the phase response function of the Kuramoto oscillators (see methods, Equation 4). The center frequency was set to 6 Hz in all of these simulations. Overall, theta oscillations in these cases are less sinusoidal and show more abrupt phase changes than in the physiological case. (**A**) A phase offset of −π/2 leads to an overall theta oscillation of 4 Hz, with a second peak following the main theta peak. (**B**) A phase offset of +π/2 reduces the peak of theta, resetting the rhythm to the middle of the ascending phase. (**C**) A phase offset of π or−π leads to the CA1 output resetting the theta rhythm to the trough of theta.

2.4. We may wonder whether a different hippocampal model of TNGO, of which there are many published (for example [1-6]) would show the same effect under neurostimulation. This seems very unlikely […][1] Hyafil A, Giraud AL, Fontolan L, Gutkin B. Neural cross-frequency coupling: connecting architectures, mechanisms, and functions. Trends in neurosciences. 2015 Nov 1;38(11):725-40.[2] Tort AB, Rotstein HG, Dugladze T, Gloveli T, Kopell NJ. On the formation of gamma-coherent cell assemblies by oriens lacunosum-moleculare interneurons in the hippocampus. Proceedings of the National Academy of Sciences. 2007 Aug 14;104(33):13490-5.[3] Neymotin SA, Lazarewicz MT, Sherif M, Contreras D, Finkel LH, Lytton WW. Ketamine disrupts theta modulation of gamma in a computer model of hippocampus. Journal of Neuroscience. 2011 Aug 10;31(32):11733-43.[4] Ponzi A, Dura-Bernal S, Migliore M. Theta-gamma phase-amplitude coupling in a hippocampal CA1 microcircuit. PLOS Computational Biology. 2023 Mar 23;19(3):e1010942.[5] Bezaire MJ, Raikov I, Burk K, Vyas D, Soltesz I. Interneuronal mechanisms of hippocampal theta oscillations in a full-scale model of the rodent CA1 circuit. Elife. 2016 Dec 23;5:e18566.[6] Chatzikalymniou AP, Gumus M, Skinner FK. Linking minimal and detailed models of CA1 microcircuits reveals how theta rhythms emerge and their frequencies controlled. Hippocampus. 2021 Sep;31(9):982-1002.

The highlighted publications, while very important in their findings regarding theta-gamma phase-amplitude coupling, focused on specific subfields of the hippocampus. In our work, we aimed to develop a model that includes the different anatomical divisions of the hippocampal formation, while still exhibiting theta-nested gamma oscillations, which is why we decided to expand the model by Aussel et al. (2018). Exploring the behavior of all these different hippocampal models under neurostimulation is beyond the scope of the current manuscript.

Nevertheless, we have added a new figure (Figure 3 – Figure Supplement 4) showing an adaptation of our modeling approach to a generic neural mass model of theta-nested gamma oscillations (Onslow et al., 2014), which illustrates the generalizability of our findings and is described in details in our response to comment 2.1. Moreover, we have further addressed the comments of the reviewers regarding bistability and phase response curves in our responses to comments 2.2 and 2.3.

Furthermore, we have added references to all 6 of these publications in the revised version of the manuscript:

L.43-50: Moreover, the modulation of gamma oscillations by the phase of theta oscillations in hippocampal circuits, a phenomenon termed theta-gamma phase-amplitude coupling (PAC), correlates with the efficacy of memory encoding and retrieval (Jensen and Colgin, 2007; Tort et al., 2009; Canolty and Knight, 2010; Axmacher et al., 2010; Fell and Axmacher, 2011; Lisman and Jensen, 2013; Lega et al., 2016). Experimental and computational work on the coupling between oscillatory rhythms has indicated that it originates from different neural architectures and correlates with a range of behavioral and cognitive functions, enabling the long-range synchronization of cortical areas and facilitating multi-item encoding in the context of memory (Hyafil et al., 2015)."

L.415-426: “In terms of neuronal cell types, we also made an important simplification by considering only basket cells as the main class of inhibitory interneuron in the whole hippocampal formation. However, it should be noted that many other types of interneurons exist in the hippocampus and have been modeled in various works with higher computational complexity (e.g., Bezaire et al., 2016; Chatzikalymniou et al., 2021). Among these various interneurons, oriens-lacunosum moleculare (OLM) neurons in the CA1 field have been shown to play a crucial role in synchronizing the activity of pyramidal neurons at gamma frequencies (Tort et al., 2007), and in generating theta-gamma PAC (e.g., Neymotin et al., 2011; Ponzi et al., 2023). Additionally, these cells may contribute to the formation of specific phase relationships within CA1 neuronal populations, through the integration between inputs from the medial septum, the EC, and CA3 (Mysin et al., 2019). Future work is needed to include more diverse cell types and detailed morphologies modeled through multiple compartments.”

2.5. […] and indeed the quiescent state itself shown by this model seems quite artificial.

We would like to clarify the fact that the “quiescent state” mentioned by the reviewer is a simply a state where the theta input is too low to induce theta-nested gamma oscillations. In this regime, neurons are active only due to the noise term in the membrane potential, which was adjusted based on Figure S3 (Figure 2 – Figure Supplement 2, shown below), at the minimal level needed to disrupt artificial synchronization in decoupled populations. For an input of 0 nA, we acknowledge that this network is indeed fully quiescent (i.e., does not show any spiking activity). However, as soon as the input increases, spontaneous spiking activity starts to appear with an average firing rate that depends on the input amplitude and is characterized by the input-frequency curves (panel A.). Please note that adding more noise could eliminate the observed quiescence in the absence of any input, but that it would not affect qualitatively the reported results.

**Author response image 10. sa3fig10:** Figure S3 (Figure 2 – Supplement 2). Cell-intrinsic spiking activity in decoupled excitatory and inhibitory populations under ramping input. (**A**) Input-Frequency (I-F) curves for excitatory cells (left panel; pyramidal neurons with ICAN) and inhibitory cells (right panel; interneurons, fast-spiking) used in the model. Above a certain tonic input (around 0.35 nA for excitatory and 0.1 nA for inhibitory neurons), neurons can spike in the gamma range. (**B**) Raster plot showing the spiking activity of excitatory (blue, NE = 1000) and inhibitory (red, NI = 100) neurons in decoupled populations under ramping input (top trace) and in the absence of noise in the membrane potential. Despite random initial conditions across neurons, oscillations emerge in both populations due to the intrinsic properties of the cells, with a frequency that is predicted by the respective IF curves (panel A.). (**C**) Similar representation as panel B. but with the addition of stochastic noise in the membrane potential of each neuron. The presence of noise disrupts the emergence of oscillations in these decoupled populations.

2.6. Some indication that particular ion channels, CAN and M are relevant is briefly provided and the work would be much improved by examining this aspect in more detail.

We thank the reviewer for acknowledging the importance of these ion channels. We have now added a new supplementary figure (Figure 5 – Figure Supplement 4), which is described in more details in our response to comment 2.2 and illustrates the role of the CAN current in the generation of theta-nested gamma oscillations following a single stimulation pulse. Moreover, we would like to stress that the impact of CAN currents in the ability of the hippocampus to generate theta-nested gamma oscillations intrinsically, i.e., in the absence of persistent external input, has already been investigated in details by a previous computational study cited in our manuscript (Giovannini F, Knauer B, Yoshida M, Buhry L. The CAN-In network: A biologically inspired model for self-sustained theta oscillations and memory maintenance in the hippocampus. Hippocampus. 2017 Apr;809 27(4):450–463).

2.7. In summary, the work would benefit from an intuitive analysis of the basic model ingredients underlying its neurostimulation response properties.

We thank the reviewer for this suggestion. By addressing the reviewer’s previous comments (reviewer 2, comments 2.1 and 2.2), which overlap partly with the first reviewer (reviewer 1, comment 3), we believe we have improved the manuscript and have provided key information related to the way the model responds to neurostimulation.

3.1. Third, while the model is fairly realistic, considerable important factors are not included and in fact, there are much more detailed hippocampal models out there (for example [5,6]). In particular, it includes only excitatory cells and a single type of inhibitory cell. This is particularly important since there are many models and experimental studies where specific cell types, for example, OLM and VIP cells, are strongly implicated in TNGO.[5] Bezaire MJ, Raikov I, Burk K, Vyas D, Soltesz I. Interneuronal mechanisms of hippocampal theta oscillations in a full-scale model of the rodent CA1 circuit. Elife. 2016 Dec 23;5:e18566.[6] Chatzikalymniou AP, Gumus M, Skinner FK. Linking minimal and detailed models of CA1 microcircuits reveals how theta rhythms emerge and their frequencies controlled. Hippocampus. 2021 Sep;31(9):982-1002.

We thank the reviewer for pointing out these interesting avenues for future studies. As indicated in previous responses (reviewer 1, comment 1; reviewer 2, comment 2.4), we have added several paragraphs to discuss these limitations, the rationale behind our simplifications, and potential improvements. In particular, we have added the following paragraphs to discuss our simplifications in terms of connectivity and cell types:

Anatomical connectivity:

L.141-150: “Biologically, GABAergic neurons from the medial septum project to the EC, CA3, and CA1 fields of the hippocampus (Toth et al., 1993; Hajós et al., 2004; Manseau et al., 2008; Hangya et al., 2009; Unal et al., 2015; Müller and Remy, 2018). Although the respective roles of these different projections are not fully understood, previous computational studies have suggested that the direct projection from the medial septum to CA1 is not essential for the production of theta in CA1 microcircuits (Mysin et al., 2019). Since our modeling of the medial septum is only used to generate a dynamic theta rhythm, we opted for a simplified representation where the medial septum projects only to the EC, which in turn drives the different subfields of the hippocampus. In our model, Kuramoto oscillators are therefore connected to the EC neurons and they receive projections from CA1 neurons (see methods for more details).”

Cell types:

L.415-426: “In terms of neuronal cell types, we also made an important simplification by considering only basket cells as the main class of inhibitory interneuron in the whole hippocampal formation. However, it should be noted that many other types of interneurons exist in the hippocampus and have been modeled in various works with higher computational complexity (e.g., Bezaire et al., 2016; Chatzikalymniou et al., 2021). Among these various interneurons, oriens-lacunosum moleculare (OLM) neurons in the CA1 field have been shown to play a crucial role in synchronizing the activity of pyramidal neurons at gamma frequencies (Tort et al., 2007), and in generating theta-gamma PAC (e.g., Neymotin et al., 2011; Ponzi et al., 2023). Additionally, these cells may contribute to the formation of specific phase relationships within CA1 neuronal populations, through the integration between inputs from the medial septum, the EC, and CA3 (Mysin et al., 2019). Future work is needed to include more diverse cell types and detailed morphologies modeled through multiple compartments.”

3.2. Other missing ingredients one may think might have a strong impact on model response to neurostimulation (in particular stimulation trains) include the well-known short-term plasticity between different hippocampal cell types and active dendritic properties.

We agree with the reviewer that plasticity mechanisms are important to include in future work, which we had already mentioned in the limitations section of the manuscript:

L.436-443: “Importantly, we did not consider learning through synaptic plasticity, even though such mechanisms could drastically modify synaptic conduction for the whole network (Borges et al., 2017). Even more interestingly, the inclusion of spike-timing-dependent plasticity would enable the investigation of stimulation protocols aimed at promoting LTP, such as theta-burst stimulation (Larson et al., 2015). This aspect would be of uttermost importance to make a link with memory encoding and retrieval processes (Axmacher et al., 2006; Tsanov et al., 2009; Jutras et al., 2013) and with neurostimulation studies for memory improvement (Titiz et al., 2017; Solomon et al., 2021).”

4. Fourth the MS model seems somewhat unsupported. It is modeled as a set of coupled oscillators that synchronize. However, there is also a phase reset mechanism included. This mechanism is important because it underlies several of the phase reset behaviors shown by the full model. However, it is not derived from experimental phase response curves of septal neurons of which there is no direct measurement. The work would benefit from the use of a more biologically validated MS model.

We would like to confirm that the phase reset mechanism is indeed at the core of using Kuramoto oscillators to model a particular system. For more details about our choice of a phase response function and the obtained results in terms of phase response curves, we refer the reader to our response to comment 2.3.

Generally speaking, we chose to use Kuramoto oscillators as it is the simplest model that can provide an oscillatory input to another system while including a phase reset mechanism. This set of oscillators was used to replace the fixed sinusoidal wave that represented theta inputs in previous models (Onslow et al., 2014; Aussel et al., 2018; Segneri et al., 2020). Kuramoto oscillators are a well-established model of synchronization in various fields of physics. They have also been used in neuroscience to model the phase reset of collective rhythms (Levnajić et al. 2010), and the effects of DBS on the basal ganglia network in Parkinson’s disease (Tass et al. 2003, Ebert et al. 2014, Weerasinghe et al. 2019).

More detailed models of the medial septum exist in the literature (e.g., Wang et al. 2002, Hajós et al. 2004) and model the GABAergic effects of the septal projections onto the hippocampal formation. However, it is not trivial to infer the connectivity parameters and the degree of innervation between the hippocampus and the medial septum. Furthermore, the claims made in our study do not necessarily depend on the nature of the projections between the two areas. Therefore, we decided to represent the medial septum in a conceptual way and focus mostly on the effects of these projections rather than replicating them in detail.

Aussel, Amélie, Laure Buhry, Louise Tyvaert, and Radu Ranta. “A Detailed Anatomical and Mathematical Model of the Hippocampal Formation for the Generation of Sharp-Wave Ripples and Theta-Nested Gamma Oscillations.” Journal of Computational Neuroscience 45, no. 3 (December 2018): 207–21. https://doi.org/10.1007/s10827-018-0704-x.

Ebert, Martin, Christian Hauptmann, and Peter A. Tass. “Coordinated Reset Stimulation in a Large-Scale Model of the STN-GPe Circuit.” Frontiers in Computational Neuroscience 8 (2014): 154. https://doi.org/10.3389/fncom.2014.00154.

Hajós, M., W.E. Hoffmann, G. Orbán, T. Kiss, and P. Érdi. “Modulation of Septo-Hippocampal θ Activity by GABAA Receptors: An Experimental and Computational Approach.” Neuroscience 126, no. 3 (January 2004): 599–610. https://doi.org/10.1016/j.neuroscience.2004.03.043.

Levnajić, Zoran, and Arkady Pikovsky. “Phase Resetting of Collective Rhythm in Ensembles of Oscillators.” Physical Review E 82, no. 5 (November 3, 2010): 056202. https://doi.org/10.1103/PhysRevE.82.056202.

Onslow, Angela C. E., Matthew W. Jones, and Rafal Bogacz. “A Canonical Circuit for Generating PhaseAmplitude Coupling.” Edited by Adriano B. L. Tort. PLoS ONE 9, no. 8 (August 19, 2014): e102591. https://doi.org/10.1371/journal.pone.0102591.

Segneri, Marco, Hongjie Bi, Simona Olmi, and Alessandro Torcini. “Theta-Nested Gamma Oscillations in Next Generation Neural Mass Models.” Frontiers in Computational Neuroscience 14 (2020). https://doi.org/10.3389/fncom.2020.00047.Tass, Peter A. “A Model of Desynchronizing Deep Brain Stimulation with a Demand-Controlled Coordinated Reset of Neural Subpopulations.” Biological Cybernetics 89, no. 2 (August 1, 2003): 81–88. https://doi.org/10.1007/s00422-003-0425-7.

Wang, Xiao-Jing. “Pacemaker Neurons for the Theta Rhythm and Their Synchronization in the Septohippocampal Reciprocal Loop.” Journal of Neurophysiology 87, no. 2 (February 1, 2002): 889–900. https://doi.org/10.1152/jn.00135.2001.

Weerasinghe, Gihan, Benoit Duchet, Hayriye Cagnan, Peter Brown, Christian Bick, and Rafal Bogacz. “Predicting the Effects of Deep Brain Stimulation Using a Reduced Coupled Oscillator Model.” PLoS Computational Biology 15, no. 8 (August 8, 2019): e1006575. https://doi.org/10.1371/journal.pcbi.1006575.